# Control of artificial membrane fusion in physiological ionic solutions beyond the limits of electroformation

Bong Kyu Kim [1,2,4], Dong-Hyun Kang [1,3,4], Junhyuk Woo [1], Wooseung Yoon [1], Hyunil Ryu[1], Kyungreem Han [1], Seok Chung [2] & Tae Song Kim [1] ✉

Membrane fusion, merging two lipid bilayers, is crucial for fabricating artificial membrane structures. Over the past 40 years, in contrast to precise and controllable membrane fusion in-vivo through specific molecules such as SNAREs, controlling the fusion in-vitro while fabricating artificial membrane structures in physiological ionic solutions without fusion proteins has been a challenge, becoming a significant obstacle to practical applications. We present an approach consisting of an electric field and a few kPa hydraulic pressure as an additional variable to physically control the fusion, enabling tuning of the shape and size of the 3D freestanding lipid bilayers in physiological ionic solutions. Mechanical model analysis reveals that pressure-induced parallel/normal tensions enhance fusion among membranes in the microwell. In-vitro peptide-membrane assay, mimicking vesicular transport via pressure-assisted fusion, and stability of 38 days with in-chip pressure control via pore size-regulated hydrogel highlight the potential for diverse biological applications.

Membrane fusion, the process by which two separate lipid bilayers merge into one, is an essential physiological process that frequently occurs in many cellular events – fertilization, neurotransmission, carcinogenesis, and viral infection of host cells[1–6]. Nevertheless, membrane fusion does not readily occur in normal circumstances due to repulsive forces between the hydrophilic polar heads of the lipid bilayers[7–9]. This is why the fusion process in-vivo requires specific molecules, such as SNAREs, synaptotagmins, and viral fusion proteins[3,4]. The primary role of these fusion proteins is to dock for proximity and lower energy barriers between proximal membranes at the appropriate time and place to allow the regulation of the fusion process (Fig. 1a top)[1,3]. This suggests that even though it is crucial to regulate biological phenomena on purpose in biotechnology, controlling membrane fusion intentionally is challenging due to the complex interplays of fusion proteins and membranes.

The difficulty in controlling membrane fusion during the fabrication of artificial membrane structures is one of the significant obstacles to extending practical applications of lipid bilayer structures. Despite various methods that have been proposed since the 1970s to produce stable lipid bilayer structures[10,11], rehydration of dried lipid stacks with electric fields, i.e., electroformation, is a representative method to produce a solvent-free lipid bilayer membrane that does not denature biological components such as membrane proteins[11–13]. Even without fusion proteins as in-vivo, the applied electric field effectively induces the initial membrane swells and their fusion due to the electro-osmotic behavior of the medium[14,15]. Since electroformation, however, grows the lipid vesicles via membrane fusion of adjacent neighbors among randomly generated uncontrollable swells, it is not easy to control the fusion process of the lipid bilayer structure without regulating sufficient proximity of membrane swells. Micro-contact printing regulates the patterned size and thickness of the dried

[1]Center for Brain Technology, Korea Institute of Science and Technology, 5, Hwarang-ro 14-gil, Seongbuk-gu, Seoul 02792, Republic of Korea. [2]Department of Mechanical Engineering, Korea University, 145 Anam-ro, Seongbuk-gu, Seoul 02841, Republic of Korea. [3]Bionics Research Center, Korea Institute of Science and Technology, 5, Hwarang-ro 14-gil, Seongbuk-gu, Seoul 02792, Republic of Korea. [4]These authors contributed equally: Bong Kyu Kim, Dong-Hyun Kang. ✉e-mail: tskim@kist.re.kr

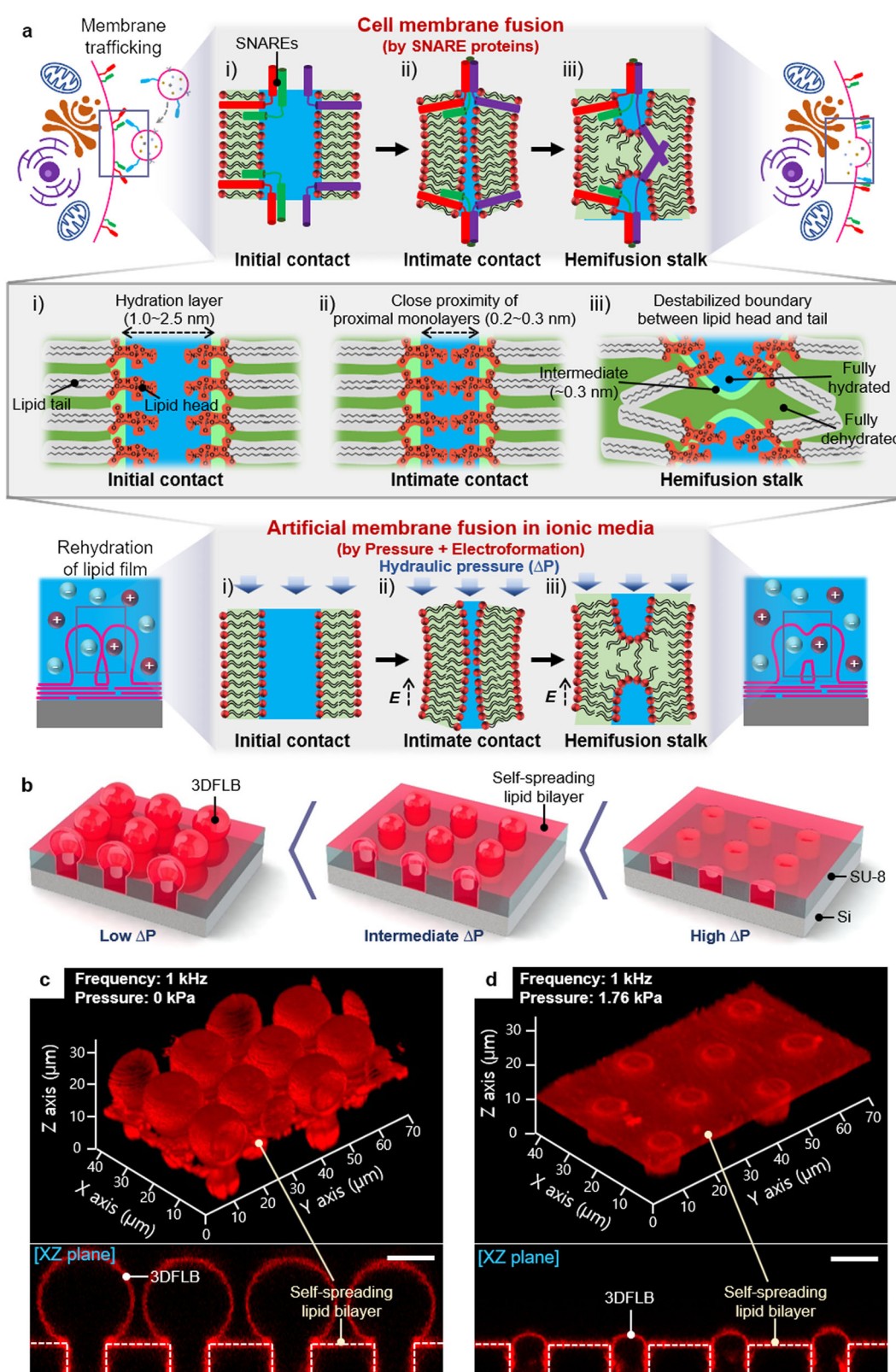

lipid stacks[16,17], thereby controlling the proximity of membrane swells, but the sealing of well-controlled giant unilamellar vesicles (GUVs) or membrane swells failed due to leaks through nanocapillaries[16,18]. The confined space of microwells promotes the fusion process by improving the proximity of controlled random swells and concentrating additional forces between the contact areas due to the expansion of membrane swells in microwells during electroformation,

allowing highly sealed 3-dimensional freestanding lipid bilayers (3DFLBs) fabrication in the non-ionic media[18] (Supplementary Fig. 1). It implies that the confined microwell acts similarly to fusion proteins in-vivo in the sense of generating proximation. However, it is still unclear whether only a confined space in the microwell under the ionic buffer condition can promote fusion and thus induce the growth of the 3DFLB array because electroformation is vulnerable to ions interfering

**Fig. 1 | 3D freestanding lipid bilayer structures (3DFLBs) generated by hydraulic pressure-assisted electroformation under physiological ionic conditions for versatile biological applications. a** Concept of artificial membrane fusion control in ionic media (physiological ionic conditions) without fusion proteins in-vitro (hydraulic pressure + electroformation) compared with cell membrane fusion (in-vivo) for membrane trafficking via SNARE proteins: i) initial contact: the apposed lipid bilayers are generally subject to the repulsive force due to the dehydration of the lipid polar heads predominating at approximately 1.0–2.5 nm intermembrane distance (hydration layer); ii) intimate contact: for membrane fusion, intimate contact between bilayers (0.2–0.3 nm) should be preceded, which is facilitated by SNAREs in cell membrane fusion and by hydraulic pressure and electric field in this study; iii) hemifusion stalk: sufficiently proximity of membranes causes destabilized boundary between the hydrophilic and hydrophobic portion of the bilayer, resulting in non-bilayer transition states which are generated that culminate in the formation of stalk and fusion pore. **b** Schematics of scalable 3DFLBs on a microwell template (SU-8 microwell on Si substrate) with diverse shapes/sizes, high stability, tight sealing under physiological ionic conditions, and biofunctionality in a solvent-free manner that involves the simultaneous application of hydraulic pressure and an electric field. Carefully controlled hydraulic pressure improves the controllability of membrane fusion under physiological ionic conditions and promotes membrane fusion, whereby hydraulic pressure modulates the shape and size of 3DFLBs from protruded spherical structures to flattened structures. **c**, **d** Reconstructed 3D confocal fluorescence microscopy images of the uniform 3DFLBs arrays fabricated at 1 kHz via electroformation without hydraulic pressure (protruded spherical structures) (**c**), and at a hydraulic pressure of 1.76 kPa (flattened structures in the microwell) (**d**). Lipid composition: DOPC with 1 mol% Rhod-PE. Scale bar: 10 μm.

with membrane swells and fusion, particularly under physiological ionic conditions[19–21].

In this work, we consider applying hydraulic pressure, as an additional variable, in addition to the passive forces exerted between the membrane swells in the confined microwell during electroformation, majorly for overcoming the facing problems of the conventional electroformation for the past 40 years. Finely regulated pressure enables lower energy barriers by membrane proximity in the same manner as fusion proteins in-vivo. We demonstrate how to control membrane fusion for fabricating 3DFLBs immobilized on the microwell array under physiological ionic conditions by applying hydraulic pressure and an electric field simultaneously in a microfluidic channel. Interestingly, when pressure and electric field are simultaneously applied, the pressure for controllable fusion decreases to a few kPa compared to 10-100 MPa of recent reports[22,23].

## Results and Discussion
### Hydraulic pressure-assisted fusion in electroformation of 3DFLBs

The proposed principal mechanism for generating 3DFLBs consists of patterning lipids into topographically defined microwell arrays (Supplementary Fig. 2) via a spin-casting and dehydrating (Supplementary Fig. 3) and rehydrating of patterned lipid stacks by aqueous solution in a microfluidic channel with AC electric fields. Simultaneously, we introduce additional pressure, fine-tuned in the microfluidic channel (Supplementary Fig. 4), to examine the interplay between confined space of microwell, electric field, and hydraulic pressure during the 3DFLBs fabrication in non-ionic media. Similar to an electric field, pressure rapidly propagates through the medium in the microfluidic channel without any time delay, allowing real-time observation of the effects of the pressure and electric field on the membrane fusion processes (Fig. 1a). Combinations of microwell, electric field, and hydraulic pressure facilitate the fabrication of 3DFLBs with various shapes and sizes in a tunable manner (Fig. 1b). With increasing applied hydraulic pressure, the shape of the 3DFLBs gradually changes from spherical to cylindrical, shrinking into the microwell, which is confirmed in the 3D fluorescence images reconstructed from the 2D slices for representative 3DFLBs fabricated at 0 kPa (Fig. 1c) and 1.76 kPa (Fig. 1d) under the same AC frequency of 1 kHz.

Figure 2a shows the time-lapse confocal images observed growth details of the 3DFLBs at the representative pressure of 1.76 kPa. We classify the whole growth period of 3DFLBs into three stages: early stage (-1800 s; rapid membrane swell and fusion), middle stage (1800 ∼ 3600 s; growth of membrane structures), and late stage (-3600 s; growth saturation and sealing). In conventional electroformation (electric field only, Supplementary Fig. 1), the membranes explosively swell and form multiple multilayer membrane swells in the early stage (Supplementary Movie 1), followed by single bilayer lipid structures in the late stage via fusion in the middle stage. In contrast, under hydraulic pressure (1.76 kPa), dried lipid stacks swell, fuse, and

form a single multilayer lipid structure in the microwell in the initial 600 s of the early stage (Supplementary Movie 2), growing in the middle stage and then finally saturating to form 3DFLBs in the late stage. It indicates that the interplay effect of the electric field and pressure dominates in the early stage when membrane swells majorly occur and coalesce.

Figure 2b is schematics explaining the effect of pressure in the microwell at the initial 600 s of the early stage. Without hydraulic pressure (conventional electroformation), only the confined space of the microwell promotes membrane fusion or merging by increased proximity due to close packing among membrane swells. On the other hand, hydraulic pressure directly acts on the membrane swells in addition to the close packing effect, resulting in an oblate deformation. Therefore, inducing an increased membrane tension as well as enhanced proximity to each other. It suggests that relatively low pressure (up to about 1.76 kPa) is sufficient to fuse or merge for a single multilayer structure within initial 600 s in the confined microwell, demonstrating the interplay effect of pressure and electric field.

It is well-known that the apposed lipid bilayers are generally subject to long-range electrostatic and van der Waals interaction with the repulsive force due to the dehydration of the lipid polar heads predominating at approximately 1.0–2.5 nm intermembrane distances (Fig. 1a)[7]. These strong hydration forces, which are exponentially dependent on the intermembrane distance, must be overcome or reduced to obtain intimate contact between bilayers for membrane fusion. For example, hemifusion of GUVs requires about 10 MPa pressure for the intimate contact between bilayers of approximately 0.2–0.3 nm[22]. Complete fusion of GUV requires 18 hrs under 100 MPa pressure during gentle hydration without an electric field[23]. In contrast, we successfully control membrane fusion that occurs quickly in the early stage of electroformation despite slight additional pressure below 1.76 kPa (Supplementary Fig. 5), reflecting the interplay effect of the confined space of microwell, the electric field, and the hydraulic pressure.

To clarify the influence of hydraulic pressure on the morphology and tension of membrane swells and 3DFLBs, we apply mechanical models based on the geometric theory of spherical membranes[24–26] (detailed description in Supplementary Methods). The membrane tension ($\tau$) acting at the interface of 3DFLBs and microwell is denoted from the vector sum of corresponding forces per unit length along the x-axis ($N_x$), the vertical shearing force acting on the force normal to the x-axis ($Q_x$), and the mending moment per unit length parallel to the y-axis ($M_y$). Hydraulic pressure induces tension, increasing nonlinearly with pressure in the microwell. The calculated membrane tension to the sidewall of the microwell is in the range of 0.83-6.29 mN/m with hydraulic pressure up to 1.76 kPa (Fig. 2c, details shown in Supplementary Figs. 19-21), which is comparable with an additional tension of 3.4-5.0 mN/m with SNARE proteins in vivo for fusion[27]. The mesoscopic simulation reports the necessity of about 10 times larger membrane tension of above 50 mN/m for effective fusion without SNARE or fusion

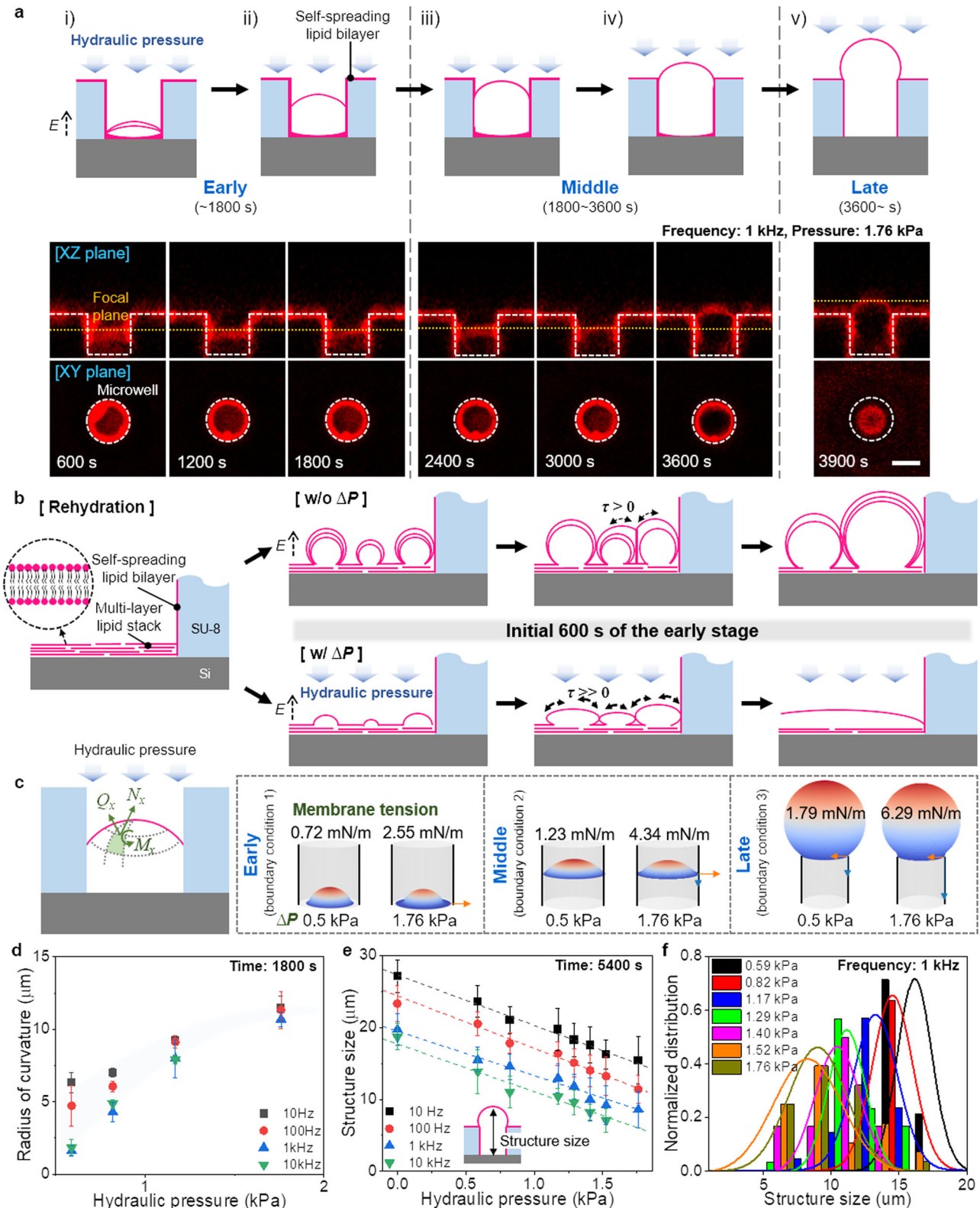

peptides[28]. We suppose that the interplay of the confined space of the microwell, electric fields, and pressure may act like SNARE or fusion peptides in vivo; thus, the additional tension generated by the hydraulic pressure promotes membrane fusion further.

The enhanced fusion induced by the applied hydraulic pressure is confirmed by the two- to fourfold change in the radius of curvature of the membrane swells formed in the microwells at different AC frequencies at the end of the early stage (-1800 s) (Fig. 2d). Interestingly,

as the pressure increases, the frequency dependence of the radius of curvature decreases, and larger pressure dependence proves that the membrane swells deform and widen under hydraulic pressure, enhancing membrane fusion.

Additionally, we can control the size of the 3DFLBs via different AC frequencies and hydraulic pressures (Fig. 2e). According to the Maxwell-Wagner model, a lipid membrane acts like an insulator[29]; thus, the charging time of the lipid membrane decreases at high AC

**Fig. 2 | Formation of 3DFLBs arrays with controllable shapes and sizes on a microwell template via hydraulic pressure-assisted electroformation.**
**a** Schematics and representative cross-sectional confocal fluorescence microscopy images of the generation of 3DFLBs in 10 mM sucrose solution: i) multilayer lipid membranes swelled after encountering solution; ii) fused single lipid membrane formed by carefully controlling the hydraulic pressure at the end of the early stage; iii-iv) development of the single membrane into a 3DFLBs in the middle stage; and v) the finalized 3DFLBs, which are robustly attached to the self-spreading lipid bilayer on the microwell surface, thus forming a tight seal, in the late stage. Scale bar: 5 μm. **b** Schematic illustrations of the initially rehydrated lipid membranes with and without hydraulic pressure. Without hydraulic pressure, the explosively swelled multilayer lipid membranes generate membrane tension (τ) only through contact with other membranes, resulting in multiple membrane swells. With hydraulic pressure, the squeezed and widened lipid membranes enhance the membrane tension, fuse with each other, and form a single lipid bilayer within

600 s. **c** Mechanical changes of 3DFLBs upon applying hydraulic pressure at different initial radii of each boundary condition. The initial radius of 3DFLBs is 3, 5, and 7 μm at the early, middle, and late stages, respectively. The divided tensions acting at the interface toward the bottom (blue arrow) and the side wall (orange arrow), respectively, denote the vector sum of $N_x$ and $Q_x$. **d** Radius of curvature of the membrane swells produced at different AC electric field frequencies as a function of the hydraulic pressure at the end of the early stage. Dependence of the radius of curvature varies on the electric field frequency at low pressure, indicating the improved controllability of the membrane swells when hydraulic pressure is applied. **e** Structure size of the finalized 3DFLBs produced different AC electric frequencies as a function of the hydraulic pressure. Data are presented as means ± standard deviation ($n = 5$ in (**d**) and $n = 55$ in (**e**) independent samples). **f** Size distributions of the finalized 3DFLBs produced with different hydraulic pressures at an AC electric field frequency of 1 kHz.

frequencies, reducing AC field effects and 3DFLBs size. Hydraulic pressure also gently modulates the size and shape of the 3DFLBs by horizontally widening the structure and suppressing growth. Notably, our method fabricates an immobilized 3DFLBs array in uniform size with a coefficient of variation of less than 23% (Fig. 2f).

## Control of membrane fusion and sealing of 3DFLBs under physiological ionic conditions

Based on the enhanced fusion by applying hydraulic pressure during electroformation in non-ionic media, we explore the effects of pressure on generating 3DFLBs arrays and their sealing in the microfluidic channel under physiological ionic conditions. For tight sealing and complete compartmentalization of the 3DFLBs, it is critical for controlling membrane fusion between the 3DFLBs and the self-spreading lipid bilayer over the entire microwell surface[18], which occurs mainly in the late stage (-3600 s) compared to early and middle stage dominating 3DFLBs generation (Fig. 3a). Conventionally, lipid membranes on solid surfaces rarely fuse with proximal lipid vesicles due to electrostatic repulsion, which becomes much severer in ionic solutions[30].

Figure 3b shows a comprehensive relationship between the shape of lipid structures generated with the variation of pressure and AC frequency and their sealing characteristics in a non-ionic media (Supplementary Fig. 6). In zone 2, the frequency-dominant region below about 0.8 kPa, 3DFLBs are successfully fabricated but leaky. The tight sealing of 3DFLBs is achieved in zone 3 by a combination of appropriate frequency and hydraulic pressure. The enhanced membrane fusion between the 3DFLBs and the self-spreading lipid bilayer via pressure-assisted electroformation extends the zone 3 to frequencies above 10 Hz which is a border of the "multilayer/multistructure" region (zone 1) and pressure below 1.76 kPa (Supplementary Fig. 7). In zone 4, 3DFLBs have negative curvatures, resulting in rupture and "no 3DFLBs" (Supplementary Figs. 19-21).

Under physiological ionic conditions, generating the 3DFLBs in regulated shape and size via electroformation is challenging. Compared to the non-ionic media conditions, as shown in Fig. 3c, each zone has significant shifts corresponding to the shape and sealing characteristics of 3DFLBs in KCl solutions (Supplementary Fig. 8). In zone 1, dramatically expanded in the KCl solution (approximately 300mOsml/L), random and spontaneous membrane swells do not fuse in the microwell and only grow individually, forming leaky multivesicular and multilayer structures (Fig. 3d). Compared to non-ionic condition, zone 2 and 3 narrow to areas above about 1 kHz. For representative conditions of 10 kHz and 0.82 kPa corresponding to zone 2, membrane swells generated at higher frequency and pressure than zone 1 can create the 3DFLBs. However, they are still leaky since fusion for sealing in the late stage requires higher pressure (Fig. 3e). We can confirm that robust 3DFLBs forms only at higher pressures sufficient to overcome repulsion in physiological KCl solutions (Fig. 3f). Despite physiological KCl solution, it is clear that only a combination of sufficiently high

hydraulic pressure and appropriate AC frequency over 500 Hz allows the fusion of membrane swells in microwell at the early stage (Supplementary Fig. 9) as well as the fusion of 3DFLBs and self-spreading lipid bilayer at the late stage (Supplementary Fig. 10). Similarly, the application of pressure can control membrane fusion in different physiological solutions, such as NaCl (Fig. 3g) and DPBS (Fig. 3h) (Supplementary Fig. 11). It is also possible to generate 3DFLBs with uniform size distributions regardless of the kind of ions in different buffers via pressure-assisted electroformation (Supplementary Fig. 12). In addition to DOPC, the synergy of microwell structures, an electric field, and hydraulic pressure enables the precise control of membrane fusion, leading to the fabrication of 3DFLBs. Various lipid compositions, including bilayer lipid (SOPC; Fig. 3i), non-bilayer lipid (POPE; Fig. 3j), a mixture of bilayer and non-bilayer lipid (DOPC:DOPE = 50:50; Fig. 3k), and even a mixture containing charged lipid (DOPC:DOPE:DOPS = 50:20:30; Fig. 3l), have all successfully produced 3DFLBs with a high level of tight sealing using the same methodology. In contrast to the conventional electroformation, we successfully achieve tightly sealed 3DFLBs with controllable shape and size by regulating the interplay between hydraulic pressure and the electric field, particularly under physiological ionic conditions.

## Biological reactions of immobilized 3DFLBs under physiological ionic conditions

Incorporating membrane proteins and peptides into artificial membranes is essential for model cell membrane studies and cell-mimicking biosensors and bioreactors[31–33]. Therefore, it is necessary to investigate the integration of proteins or peptides into the 3DFLBs generated under controlled pressure and AC frequency in physiological solutions. A pore-forming peptide, melittin, the primary component of bee venom, is a potent cytolytic anticancer peptide involved in pathological and physiological responses[34–36]. It binds within milliseconds to lipid membranes and adopts an amphipathic α-helical conformation, oriented either parallel (inactive) or perpendicular (active; pore formation) to the membrane[35,36] (Fig. 4a). After pre-confirming tight sealing of 3DFLBs, we infuse Alexa Fluor 488 (green fluorescence) and melittin in a microfluidic channel to evaluate melittin pore formation and confirm bilayer membrane (Fig. 4b; Supplementary Fig. 13). Figure 4c shows the fluorescent intensity variation inside the 3DFLBs depending on melittin concentration of 0, 3, and 10 μM. As the melittin concentration increases, the transport onset time of dye molecules becomes faster because the number of melittin monomers to be bound and the pore-forming rate increases, which confirms the unilamellar and bio-functionality of the 3DFLBs generated under controlled pressure and AC frequency.

Vesicular transport or endocytosis involving membrane fusion is a significant cellular activity responsible for molecular transport between various membrane-enclosed compartments[37,38]. We mimic vesicular transport by delivering LUVs (approximately 100 nm)

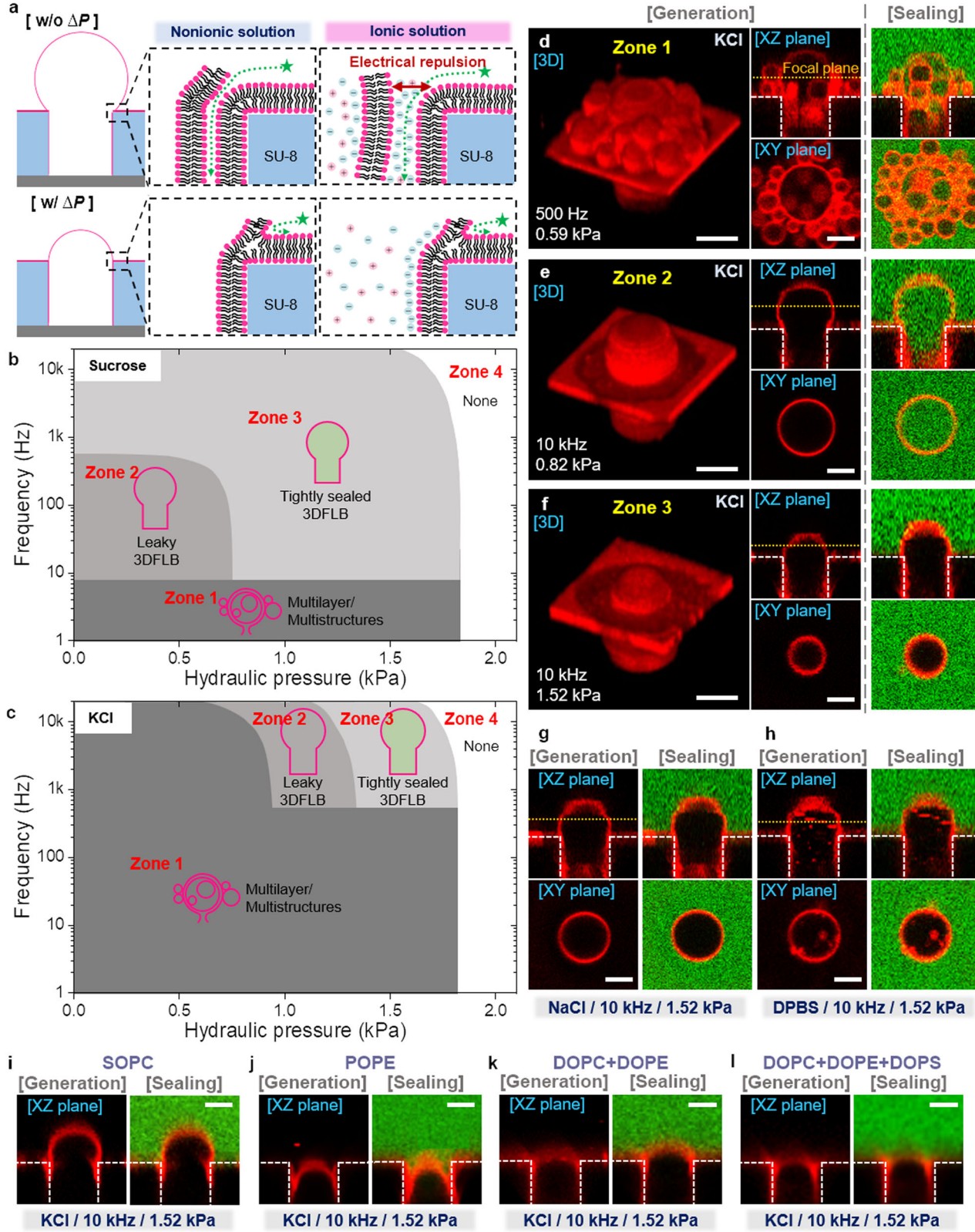

containing a fluorescent dye into the immobilized 3DFLBs in a microfluidic channel. Membrane fusion for vesicular transport typically requires biomembrane components such as SNARE or fusion peptides[5,38]. Instead, we introduce a pressure for regulating membrane fusion (Fig. 4d). By pressure, infused LUVs are sequentially fused into the 3DFLBs, enriching the green fluorescence intensity in sealed 3DFLBs (Fig. 4e; Supplementary Fig. 14). The fluorescent intensity of

the 3DFLBs increases rapidly with pressure compared to that in the absence of pressure and then remains constant after washing the outer LUVs (Fig. 4f). In general, although LUV-GUV adhesion works well compared to GUV-GUV adhesion, LUV-GUV rarely fuses, which supports that applied pressure promotes the fusion of the bound LUVs[39]. Given the importance of incorporating various cellular components or membrane proteins to fabricate synthetic cells or membranes while

**Fig. 3 | Tightly sealed 3DFLBs in sucrose and physiological ionic solutions.**
**a** Schematics of the tight sealing mechanism via hydraulic pressure in sucrose (non-ionic) and physiological (ionic) solutions. Outside molecules move into the unsealed 3DFLBs fabricated without hydraulic pressure through nanocapillaries. Hydraulic pressure promotes tight sealing of the 3DFLBs even under physiological ionic conditions, in which the electrical repulsion between lipid membranes is enhanced. **b, c** Morphological and hermetical phase diagrams of the 3DFLBs formed at different AC frequencies and hydraulic pressures in 10 mM sucrose solution (**b**) and KCl solution (140 mM KCl, 10 mM HEPES; 300 mOsml/L) (**c**). **d–f** Representative 3D reconstructed and cross-sectional confocal fluorescence microscopy images of 3DFLBs corresponding to each zone in (**c**) and the corresponding sealing test: multilayer/multistructure formed at 500 Hz frequency and 0.59 kPa hydraulic pressure (Zone 1) (**d**), leaky 3DFLBs formed at 10 kHz and 0.82 kPa (Zone 2) (**e**) and tightly sealed 3DFLBs formed at 10 kHz and 1.52 kPa (Zone 3) (**f**) in KCl solution. Lipid composition: DOPC

with 1 mol% Rhod-PE. **g, h** Representative cross-sectional confocal fluorescence microscopy images and the corresponding sealing tests of the tightly sealed 3DFLBs generated in NaCl solution (140 mM NaCl, 10 mM HEPES) (**g**) and 150 mM DPBS solution (both solutions are physiologically ionic at 300 mOsml/L) (**h**) at 10 kHz frequency and 1.52 kPa hydraulic pressure. Lipid composition: DOPC with 1 mol% Rhod-PE. **i–l** Representative cross-sectional confocal fluorescence microscopy images and corresponding tight-sealing tests of 3DFLBs generated in KCl solution with different lipid compositions as SOPC (**i**), POPE (**j**), a mixture of DOPC (50 mol%) and DOPE (50 mol%) (**k**), and a mixture of DOPC (50 mol%), DOPE (20 mol%), and DOPS (30 mol%) (**l**). All the lipids are labeled with 1 mol% Rhod-PE. The 3DFLBs with various lipids were generated under the same conditions as 10 kHz frequency and a hydraulic pressure of 1.52 kPa. All the representative 3D reconstructed and cross-sectional confocal fluorescence microscopy images of 3DFLBs were obtained from repeated experiments more than 4 times and their scale bar is 5 μm.

enhancing system complexity, pressure may be a candidate option for controlling membrane fusion to mimic cell activity.

### In-chip pressure control and improved stability of 3DFLBs
By simultaneously controlling the hydraulic pressure and electric field, we successfully fabricate 3DFLBs with sturdy structures. However, the hydraulic pressure controlled with external equipment leads to potential disturbances, degrading the stability and yield in 3DFLBs generation. Therefore, instead of external pressure regulation, we test an internal pressure generation using a hydrogel block overlaid on the microwell array in the microfluidic channel to produce additional pressure in the microwell (Fig. 5a). Figure 5b represents confocal microscopy images of the hydrogel block-based 3DFLBs comparable to hydraulic pressure of 1.76 kPa, proving that a hydrogel block can replace the hydraulic pressure controlled by external equipment in the microfluidic channel. Since the generated pressure by the hydrogel block depends on the size of the hydrogel pores, we adjusted the pore size by regulating the mixing ratio of the two types of hydrogels with 15:0, 10:5, and 5:10 wt% PEGDMA 1000 and PEGDMA 3400, respectively (Fig. 5c). With the PEGDMA 1000-only hydrogel block, the growth of 3DFLBs stops below the opening of the microwell due to high pressure. In contrast, a large pore hydrogel mixture applies relatively low pressure, causing the 3DFLBs to grow and spread out over the microwell opening under the overlaid hydrogel. As the mixing ratio of the PEGDMA 3400 increases, the pore diameter of the hydrogel increases from 4.27 to 19.24 μm, resulting in reduced pressure (Fig. 5d), and therefore increases the structure size of 3DFLBs (Fig. 5e). The hydrogel block enhances the uniformity of the 3DFLBs to a coefficient of variation of less than 7.69% and also significantly improves the stability of 3DFLBs. (Fig. 5f; Supplementary Fig. 15). Compared with the previous result of the 3DFLBs of 5.5 days[18], 3DFLBs fabricated with hydraulic pressure are stable for at least 17 days. Moreover, the hydrogel block that prevents external flow disturbance increases the stability to 38 days (Fig. 5g). This enhanced stability of the lipid bilayer structure could pave various applications such as artificial cells, cell-mimetic biosensors, and bioreactors not only in the laboratory but also in commercial applications.

In summary, beyond the limits of conventional electroformation methods over the past 40 years, this study presents a systematic approach to overcome the limitations of conventional electroformation in fabricating tightly sealed GUVs (3DFLBs) array immobilized on substrates under physiological ionic conditions by simultaneously modulating pressure and electric field. The combination of a confined space of microwell, electric field, and hydraulic pressure enables the formation of lipid bilayer structures with controlled monodisperse size and shape, remarkable stability, and bio-functionality, opening up new possibilities for their practical implementation in biotechnology and pharmaceuticals. Moreover, this study can be further improved by integrating electrodes into the microwell template or incorporating

various membrane proteins and biomolecules, thereby further expanding its applications.

## Methods
### Materials
Si wafers (p-type, 100) were purchased from Hissan (South Korea), and SU-8 2050, SU-8 3010, and SU-8 developers were purchased from Kayaku Advanced Materials, Inc. (USA). All lipids used, namely, 1,2-dioleoyl-*sn*-glycero-3-phosphocholine (DOPC), 1-stearoyl-2-oleoyl-*sn*-glycero-3-phosphocholine(SOPC), 1,2-dioleoyl-*sn*-glycero-3-phosphoethanolamine (DOPE), 1-palmitoyl-2-oleoyl-*sn*-glycero-3-phosphoethanolamine (POPE), 1,2-dioleoyl-*sn*-glycero-3-phospho-L-serine (DOPS), 1,2-dioleoyl-*sn*-glycero-3-phosphoethanolamine-*N*-(lissamine rhodamine B sulfonyl) (Rhod-PE), and 1,2-dioleoyl-*sn*-glycero-3-phosphoethanolamine-*N*-(7-nitro-2-1,3-benzoxadiazol-4-yl) (18:1 NBD-PE), were purchased from Avanti Polar Lipids, Inc. (USA). Poly-dimethylsiloxane (PDMS) was purchased from Dow Corning (USA) and used by mixing the base and curing agent at a ratio of 10:1. Poly(ethylene glycol) diacrylate (PEGDMA; Mn: 1000 and 3400) and Alexa Fluor 488 dye (succinimidyl ester) were purchased from Thermo Fisher Scientific, Inc. (USA). 2-Hydroxy-2-methyl-propiophenone (hydrogel curing agent), poly(ethylene glycol)-block-poly(propylene glycol)-block-poly(ethylene glycol) (Poloxamer 188) solution, 4-(2-hydroxyethyl)−1-piperazineethanesulfonic acid (HEPES), sucrose, potassium, sodium, chlorine, Dulbecco's phosphate buffered saline (DPBS), melittin, and all solvents were purchased from Sigma–Aldrich (USA) and used as received.

### Fabrication of microwell array on a silicon substrate
We fabricated a microwell array on a silicon substrate using the epoxy-based negative photoresist SU-8. The Si wafers were cleaned with acetone and isopropyl alcohol (IPA) and rinsed thoroughly with deionized (DI) water. SU-8 was spin-coated on a Si wafer at a thickness of 8 μm. After soft baking at 65 °C for 1 min and 95 °C for 5 min, the SU-8 was exposed to UV light using a mask aligner (MA6 Mask Aligner, Suss MicroTec, Germany), followed by postexposure baking at 65 °C for 1 min and 95 °C for 2 min, developing in a fresh SU-8 developer with agitation, rinsing with IPA and DI water, and hard baking at 110 °C for 2 min.

### Patterning of lipids in the microwell array
An appropriate amount of chloroform was added to a vial containing lipid or lipid mixture with 1 mol% Rhod-PE for fluorescence observation to obtain 17 mM lipid solutions. The lipid we used is DOPC, SOPC, or POPE. The lipid mixture we used consisted of 50 mol% DOPC and 50 mol% DOPE or 50 mol% DOPC, 20 mol% DOPE, and 30 mol% DOPS. After the chloroform was evaporated with Ar, the vial was placed under vacuum for at least 120 min to obtain a dried lipid film. The dried lipid film was hydrated with DI water under gentle agitation. Suspensions of the hydrated lipid film that contained

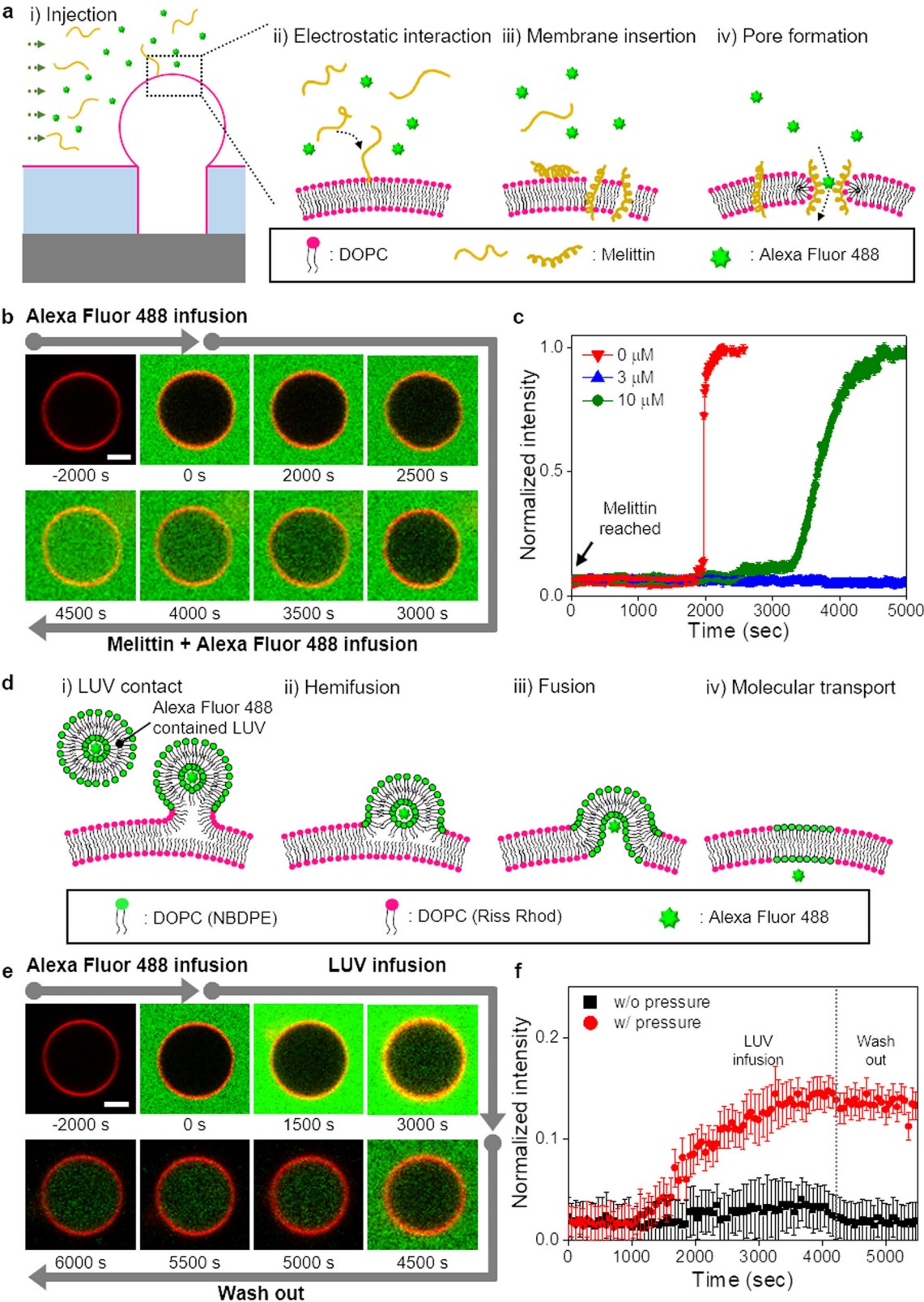

multilamellar, micrometer-sized vesicles were obtained, followed by extrusion 21 times through a 50 nm polycarbonate membrane filter (Nuclepore track-etched membranes, Whatman, Inc., UK) using an extruder (Avanti Polar Lipids, Inc., USA). The SUV dispersion was stored at 4 °C for future use. The prepared SUV dispersion was spin-coated on the SU-8 microwell templates, which were cleaned with IPA and DI water at a rate of 100 rpm for 30 s and 1000 rpm for 30 s,

followed by drying in a freeze dryer for more than 6 h to remove any traces of the aqueous solution.

## Formation of a 3DFLBs via hydraulic pressure-assisted electroformation

We fabricated simple microreactor-embedded straight microchannels for simultaneous 3DFLBs generation, observation, and utilization

**Fig. 4 | Two representative biological membrane reactions based on 3DFLBs as an application platform. a** The pathological and physiological responses with the incorporation of a pore-forming peptide (melittin) in the 3DFLBs: i) injection of melittin; ii) approach of melittin to the 3DFLBs; iii) binding and insertion of melittin into the 3DFLBs; and iv) pore formation of perpendicularly oriented melittin and inflow of fluorescent dye (Alexa Fluor 488). **b** Fluorescence microscopy images over time: infusion of Alexa Fluor 488 to confirm the sealing of 3DFLBs and infusion of a mixed solution of Alexa Fluor 488 and melittin to confirm melittin pore formation and bilayer membrane structure of the 3DFLBs. Lipid composition: DOPC with 1 mol% Rhod-PE. Scale bar: 5 μm. **c** Time-dependent influx of fluorescent dye molecules into the 3DFLBs through melittin pores at different melittin concentrations (0, 3, 10 μM) by monitoring the normalized fluorescence intensity. Data are presented as means ± standard deviation (*n* = 6 independent samples). **d** Schematic illustration representing the fusion of LUVs into the 3DFLBs as a host compartmentalized artificial cell for mimetic vesicular transport via hydraulic pressure: i)

contact of 3DFLBs and infused NBD-PE (1 mol%) labeled LUVs containing Alexa Fluor 488 (5 μM); ii) hemifusion of LUVs and 3DFLBs, which is promoted by hydraulic pressure; and iii-iv) fusion of LUVs and 3DFLBs and penetration of Alexa Fluor 488 into the 3DFLBs. **e** Fluorescence microscopy images over time: infusion of Alexa Fluor 488 to confirm the sealing of the 3DFLBs; infusion of LUVs to mimic vesicular transport; and washing out process to evaluate the fusion of LUVs and 3DFLBs by monitoring the fluorescence intensity inside the 3DFLBs. Lipid composition: (3DFLB) DOPC with 1 mol% Rhod-PE, (LUV) DOPC with 1 mol% NBD-PE. Scale bar: 5 μm. **f** Comparison of time-dependent vesicular transport with and without hydraulic pressure by monitoring the normalized fluorescence intensity inside the 3DFLBs. The slight decrease in the fluorescence intensity after washing out the remaining LUVs is due to the removal of background noise due to the infused LUVs. Data are presented as means ± standard deviation (*n* = 6 independent samples).

using PDMS soft lithography technology (detailed structure shown in Supplementary Fig. 4a, top). Briefly, the bottom part of the microchannel was prepared by bonding a PDMS layer (65 × 20 × 0.5 mm³) with an open area (15 × 15 mm²) for Si template insertion. The top part of the microchannel was prepared by integrating a PDMS layer (65 × 20 × 0.5 mm³) with an indium tin oxide (ITO, 200 nm thick, ~50 Ω/cm)-coated slide glass (65 × 25 mm²) and the microchannels (150 μm thick). For the experiments, the lipid-coated SU-8 microwell template was placed in the open area on the bottom part, and the top and bottom surfaces of the ITO and SU-8 microwell templates facing each other were bonded together. The assembled top and bottom parts were placed between in-house-built PMMA housing with screws for tight coupling. Four hydration solutions were used in these experiments: sucrose solution (10 mM sucrose), KCl solution (140 mM KCl, 10 mM HEPES), NaCl solution (140 mM NaCl, 10 mM HEPES), and DPBS solution (150 mM DPBS). Upon infusion of a hydration solution into the microchannel at a rate of 40 μL/h using a syringe pump (NE-1000, New Era Pump Systems Inc., USA), a sinusoidal alternating current (AC) electric field was applied between the ITO and the Si substrate under the SU-8 microwell with a function generator (33210 A, Keysight, USA). Hydraulic pressure was applied by varying the height of the syringe pump (details shown in Supplementary Fig. 4a). The electric field with a specific frequency of 10 Hz to 10 kHz was increased by 50 mV every 5 min until it reached 600 mV, and then the electric field was maintained at that level for more than 30 min to form and seal the 3DFLBs. After the 3DFLBs were fully grown throughout the microwells via hydraulic pressure-assisted electroformation, the AC electric field was turned off, and a hydration solution containing a fluorescent dye (3 μM Alexa Fluor 488) was injected into the microchannel to confirm the sealing of the 3DFLBs.

### Evaluation of hydraulic pressure in the microchannel based on the height of the syringe pump

An air compression-based pressure measurement method was applied to directly measure the hydraulic pressure in the microchannel based on the height of an external syringe pump. Static pressure measurements were difficult to obtain due to the limited space in the microchannel. Therefore, we used a simple, cost-effective method in which a pressure tap with a closed end and a small width and height was integrated along the length of the microchannel and used as a pressure sensor. Additional details about this approach can be found in a previous study[40].

The mathematical model we used for determining the pressure is based on the ideal gas law and the interfacial pressure difference from surface tension within the microchannel. The pressure of the trapped air can be expressed as

$$P_{air} = \frac{m_{air}RT}{V_{compressed}} \tag{1}$$

where $P_{air}$, $\rho$, $R$, $T$, $m_{air}$, and $V_{compressed}$ are the air pressure, air density, gas constant, temperature, air mass in the pressure tap, and air volume in the pressure tap, respectively. In the steady state, the local pressure ($P$) balances the trapped air pressure and interfacial pressure difference. The pressure difference ($P_{diff}$) across the water–air interface is given by

$$P_{diff} = \gamma\left(\frac{1}{R_1} + \frac{1}{R_2}\right) = \gamma\left(\frac{2\cos\theta_1}{W} + \frac{2\cos\theta_2}{H}\right) \tag{2}$$

where $\gamma$ is the surface tension, $R_1$ and $R_2$ are the principal radii of curvature of the interface, and $W$ and $H$ are the width and height of the pressure tap, respectively. The contact angle $\theta$ is measured at the solid–liquid contact. Therefore, $P_{water}$ can be formulated as

$$P = P_{air} - P_{diff} = \frac{m_{air}RT}{V_{compressed}} - \gamma\left(\frac{1}{R_1} + \frac{1}{R_2}\right) \tag{3}$$

We designed and fabricated a microchannel with two pressure taps (Supplementary Fig. 4b). Then, we calculated the water pressure with the above equation based on the measured *h* (displacement of the meniscus in the pressure taps) at various heights of the syringe pump (Supplementary Fig. 4c and d).

### Incorporation of melittin on the 3DFLBs

Before the experiment, the microchannel was treated with 3% poloxamer 188 solution for 12 h to prevent the nonspecific adsorption of peptides. Melittin monomers were dissolved in KCl solution containing fluorescent dye (3 μM Alexa Fluor 488) at concentrations of 0, 3, and 10 μM. After confirmation of the tight sealing of the 3DFLBs using KCl solution containing fluorescent dye (3 μM Alexa Fluor 488), melittin-dissolved KCl solution was infused into the 3DFLBs through a microchannel at a flow rate of 40 μL/h. The passive transport of the fluorescent dye across the melittin pore was observed over time and confirmed by measuring the fluorescence intensity inside the 3DFLBs via confocal microscopy.

### Mimetics of vesicular transport

To observe the fluorescence effect in vesicles other than the 3DFLBs, a lipid solution was produced by adding an appropriate amount of chloroform to a vial containing 10 mg/mL DOPC with 1 mol% NBD-PE. After evaporating the chloroform with Ar gas flow, the vial was placed under vacuum for at least 120 min to obtain a dried lipid film. The dried lipid film was hydrated with 1 mL KCl solution (140 mM KCl, 10 mM HEPES) containing fluorescent dye (5 μM Alexa Fluor 488) under gentle agitation, followed by extrusion through a 100 nm polycarbonate membrane filter (Nuclepore track-etched membranes, Whatman, Inc., UK) 21 times using an extruder (Avanti Polar Lipids, Inc., USA) to obtain an LUV dispersion to study the mimetics of molecular transport. After confirmation of the tight sealing of 3DFLBs using KCl solution (140 mM

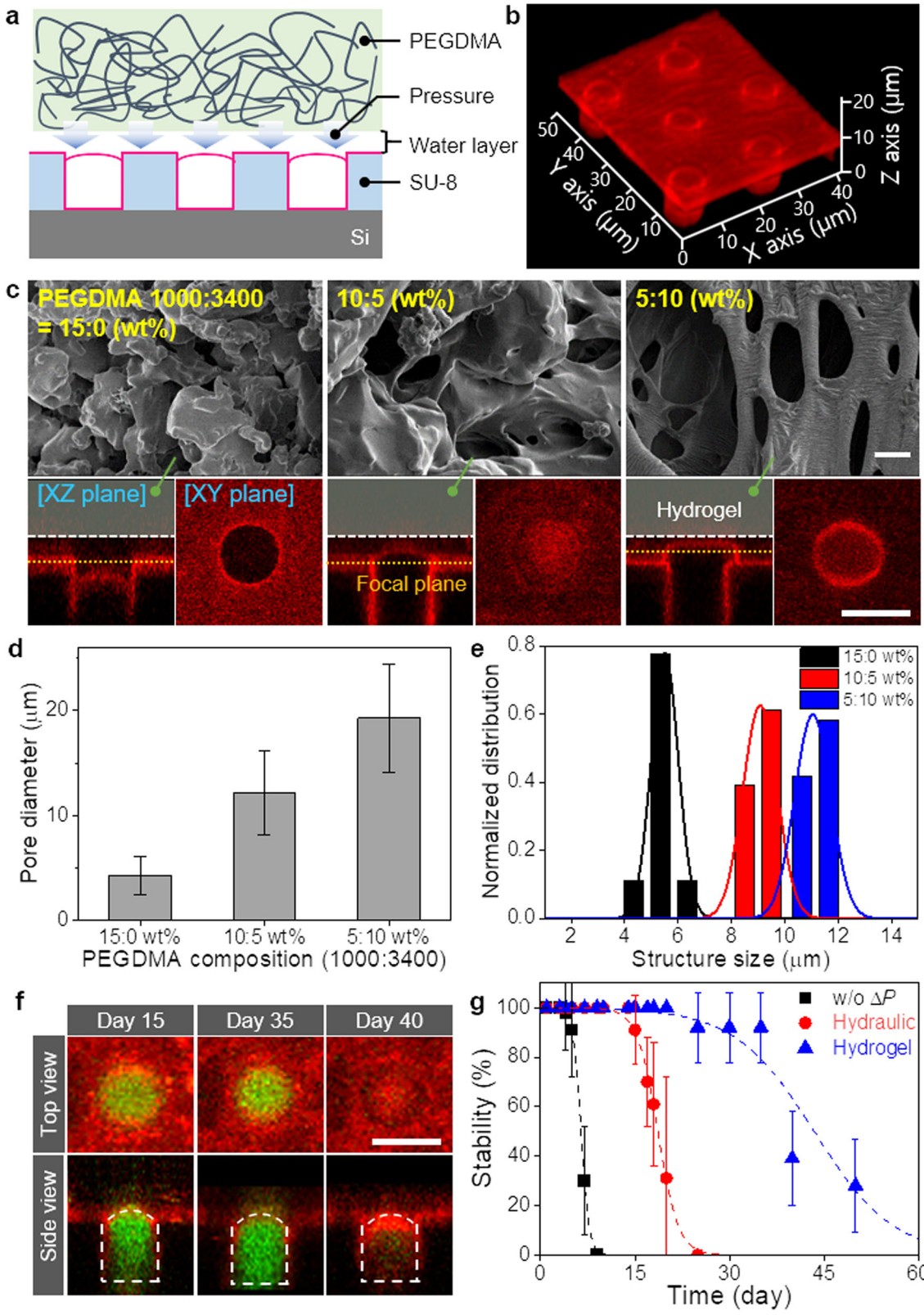

KCl, 10 mM HEPES) containing fluorescent dye (3 μM Alexa Fluor 488), the LUV dispersion was infused into the 3DFLBs through a microchannel at a flow rate of 40 μL/h. After reacting for approximately 1 h, the LUV dispersion was exchanged with KCl solution for 30 min to wash out the remaining LUVs and fluorescent dye. Identical procedures were carried out at different hydraulic pressures (0 or 1.57 kPa). The vesicular transport due to the membrane fusion between the LUVs

and 3DFLBs was observed over time and confirmed by measuring the fluorescence intensity of the 3DFLBs via confocal microscopy.

## In-chip controlled pressure-assisted electroformation via a hydrogel block
Appropriate amounts of PEGDMA 1000 and PEGDMA 3400 were dissolved in 10 mM sucrose solution to obtain 15:0, 10:5, and 5:10 wt%

**Fig. 5 | In-chip pressure control through a hydrogel block overlaid on the microwell template. a** Schematic illustration of pressure control via a hydrogel block overlaid on the microwell template. The water gap between the microwell template and the overlaid hydrogel block is approximately 3 μm. **b** 3D reconstructed confocal fluorescence microscopy image of the hydrogel block-generated 3DFLBs array (PEGDMA 1000:3400 = 10:5 wt%) at an AC frequency of 1 kHz. Lipid composition: DOPC with 1 mol% Rhod-PE. **c** SEM images of the hydrogel block and cross-sectional confocal fluorescence microscopy images of the 3DFLBs generated at an AC frequency of 1 kHz in different hydrogel compositions of PEGDMA 1000:3400 for (left) 15:0, (center) 10:5, and (right) 5:10 wt%. Lipid composition: DOPC with 1 mol% Rhod-PE. Scale bar: 10 μm. **d** Pore diameters of PEGDMA

hydrogel blocks with different compositions. Data are presented as means ± standard deviation (*n* = 10 independent samples). **e** Size distributions of 3DFLBs generated under PEGDMA hydrogel blocks with different compositions at an AC frequency of 1 kHz. **f** Time-dependent cross-sectional confocal fluorescence microscopy images of enclosed 3DFLBs containing Alexa Fluor 488 produced with a hydrogel block (PEGDMA 1000:3400 = 10:5 wt%) to verify the seal over time. Lipid composition: DOPC with 1 mol% Rhod-PE. Scale bar: 10 μm. The representative cross-sectional confocal fluorescence microscopy images of 3DFLBs were obtained from 8 times of repeated experiments. **g** Stability of 3DFLBs produced without pressure, with hydraulic pressure, and with a hydrogel block in a chip. Data are presented as means ± standard deviation (*n* = 8 independent samples).

mixtures of PEGDMA 1000 and PEGDMA 3400. After 1% v/v of the coupling agent (2-hydroxy-2-methyl-propiophenone) was added to the hydrogel solution, the mixed solution was dropped on a glass slide with a spacer (150 μm thickness) and covered with another glass slide. The hydrogel solution sandwiched between the two glass slides with a spacer was cured by irradiating the solution with UV light with a wavelength of 365 nm for 40 s with a UV lamp (UVITEC, 15 W, UK). The cured hydrogel was immersed and washed in DI water 3 times to remove unknown ions. Then, the hydrogel was cut into $15 \times 15$ mm$^2$ blocks. The cut hydrogel block was overlaid on the microwell template before assembly in the microchannel. Then, the 3DFLBs were produced in the same manner as in the experiments without external hydraulic pressure.

### Imaging and characterization

To observe and analyze the 3DFLBs, we generated 3DFLBs in a reactor-embedded microchannel directly mounted on the stage of a confocal microscope (LSM 700 confocal microscope equipped with a ×40 C-Apochromat (numerical aperture 1.2), ZEISS, Germany) or a fluorescence microscope (Zeiss LSM 5 PASCAL Axioplan 2 microscope equipped with appropriate filter sets and ×10 Epiplan-Neofluar (numerical aperture 0.30), ×20 LD Epiplan (numerical aperture 0.40), and ×50 LD Epiplan (numerical aperture 0.50) objectives, ZEISS, Germany). The images were captured using a Retiga6000 CCD camera (Qimaging, Canada) and Qimaging Pro software (version 7.0). Scanning electron microscopy (SEM; Nova Nano SEM 200, FEI, USA) measurements of the SU-8 microwell templates were performed to determine the dimensions of the microwells.

### Reporting summary

Further information on research design is available in the Nature Portfolio Reporting Summary linked to this article.

## Data availability

All data supporting the findings and conclusions of this study are available within the paper and its Supplementary Information files. All other relevant data are available from the corresponding author upon request. Source data are provided with this paper.

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

## Acknowledgements
This work was supported by the National Research Foundation of Korea (NRF) grant funded by the Korea government (MIST) (NRF-2020R1A2C2100363 to T.S.K.), and also supported by KIST Institutional Program (2E32911 to T.S.K. and 2E32921 to K.H.).

## Author contributions
T.S.K. and D.-H.K. conceived and designed the research. B.K.K. and D.-H.K. designed and performed all the experiments. H.R. designed the experiments for pressure-assisted electroformation with a hydrogel block and discussed and commented on the paper. K.H., J.W., and W.Y. conceptualized and performed the calculation and analysis of the mechanical model of 3DFLBs. S.C. supported the experiments and commented on the paper. B.K.K., D.-H.K., K.H., and T.S.K. co-wrote the manuscript.

## Competing interests
The authors declare no competing interests.
