## [Peer Review File · Nature Communications]

Control of artificial membrane fusion in physiological ionic solutions beyond the limits of electroformationReviewers' comments:

Reviewer #1 (Remarks to the Author):

Manuscript: Control of membrane fusion, beyond the limits of electroformation by Bong Kyu Kim and colleagues

The paper addresses an important and timely technological problem. The authors present an alternative electroformation protocol by implementing an electric field with hydraulic pressure to form giant-unilamellar vesicles in microwells. The method is interesting and it is important that we better understand the fusion and growth of vesicles, but the presented data are rather preliminary and do not address key issues in the field. The challenge is not so much to form vesicles from DOPC lipid but to form vesicles from physiologically relevant lipid mixtures with integral membrane proteins embedded in the lipid bilayer and to encapsulate components reproducibly and with high efficiency. The present paper does not address these issues.

1. The statement in the introduction that "... electroformation is an almost sole and representative method to produce solvent-free lipid bilayer membrane ..." is not correct. There are various alternative methods such as gel-assisted swelling that offer an alternative but, like electroformation, has limitations.
2. The advantage of the presented method over existing protocols is not very clear, except that one has an extra level of control by varying the pressure. Making vesicles by evaporating chloroform and hydrating DOPC lipids is relatively trivial and protocols are available to do this at physiological salt concentrations. It is more challenging to make vesicles from physiologically relevant amphiphile mixtures, including anionic and non-bilayer type of lipids and sterols, which are not tested in this paper. How does the method perform with other lipid mixtures.
3. Even more challenging is to form micrometer-size vesicles with genuine membrane proteins embedded in the bilayer. This requires forming vesicles from solvent-free membranes in which proteins have been incorporated. Again, protocols are available for making membrane protein-containing giant-unilamellar vesicles, but the efficiency is low and the variability is high. Any new method that addresses these problems is very welcome. The paper would gain enormous value if the functioning and quantitative analysis of real membrane receptors, channels and enzymes would have been tested. The autoinsertion of peptides and the pore formation by melittin (Figure 4) is not a real advance.
4. Why are 50 nm SUVs used as starting materials? Would the method work with larger vesicles (extrusion through 100, 200 or 400 nm polycarbonate filters), which are more commonly used in functional studies and allow more protein to be incorporated? Is the high curvature of SUVs important for fusion of with the 3DFLBSs? How many SUVs fuse? What is the experimental variation? Can quantitative data be presented?

5. The comparison of the electric field-hydraulic pressure method to form vesicles with SNARE-based fusion is far-fetched and not very relevant for the here presented protocol.

6. The authors mention the formation of tightly-sealed vesicle as a major advantage of the new method over conventional electroformation. However, electroformation also yields well-sealed vesicles that can be used to form and maintain electrochemical ion gradients. What is different here?

Minor points:

- The 3DFLBS is not explained and not common terminology to present vesicles. Are 3DFLBS different from giant-unilamellar vesicles?

- What is DPBS solution?

Reviewer #2 (Remarks to the Author):

In this contribution the authors describe the swelling of freestanding bilayers through fusion controlled by electric field and hydraulic pressure in different media/conditions. A lipid film is formed on a chip with an array of 8µm-diameter microwells and application of an AC field and hydraulic pressure causes the swelling of freestanding bilayers, whose size/shape depend on field and pressure parameters, as well as ionic concentration. In the absence of hydraulic pressure, swelling readily occurs giving rise to large and multilayered structures. But, under certain conditions of AC field, hydraulic pressure and medium composition, a well-defined and tightly sealed 3D freestanding bilayer is created on top of every microwell. Apart from defining the conditions for a controlled fusion-mediated swell, some bioinspired applications are shown, such as interaction with a lytic peptide, fusion-mediated vesicular transport and long-term stability.

The work is nicely done and provides a high-quality and comprehensive description of the swelling of freestanding bilayers via fusion of adjacent lamellae as a function of electric field frequency, hydraulic pressure and medium composition. The results are robust and the method gives rise to very well-defined structures modulated by the microchip array and the above-mentioned parameters. My main comment/concern regards the main purpose of the work and, as a consequence, the interest/applicability to the broad community of this very high standard journal. It was not clear to me whether the main purpose of the work was to describe a well-controlled way to provide 3D freestanding bilayers so that others could reproduce the method and apply to their own interest, or whether the main goal was to study the fusion process under the conditions explored. Either way, the work and its applicability/relevance remain restricted to the use of a very specific combination of setups, such as the microwell chip, the hydraulic pressure control and the coupled observation system. The method for obtaining the freestanding bilayers on the microwell platform was already published by the same group (ref. 18), and the current work adds the hydraulic pressure which enhances fusion of the swelled

structures. On the other hand, the relevance of the used parameters to the fusion process are very specific for the current system, and might not be general enough to apply in other situations.

Specific issues:

- Add the section "Introduction" after the abstract.
- The abbreviation 3DFLBS is only explained in the caption of Figure 1, but the whole name should appear the first time the abbreviation is used in the Introduction. The choice of the abbreviation might not be the best as well, since it is too long and not intuitive of its meaning. In previous publications the authors preferred 3D FLBs and 3DFLB.
- Small liposomes are always called SUVs, irrespective of their size. In a section they are said to have 100 nm and in another 200 nm. Yet in the methods a 50 nm-pore membrane is used in the extrusion process. Conventionally, 100nm and 200nm liposomes are called LUVs, and SUVs usually have 30-50nm diameter. Additionally, even when using a 50nm-pore membrane for extrusion, usually vesicles closer to 100nm diameter are obtained. SUVs are usually formed using a tip sonicator. So I guess all liposomes described are rather LUVs.
- In the experiment mimicking vesicular transfer via fusion, specify the types of fluorophores in the LUVs: The lipid membrane is labeled with 1 mol% NBD-PE and the LUVs encapsulate the aqueous soluble probe Alexa 488. The phrase "SUVs containing Alexa Fluor 488 and NBD-PE in the membrane" does not completely specify the system.
- Related to the previous comment, why don't we see clear evidence of lipid mixing in Fig. 4? Wouldn't we expect to have some green fluorescence on the membrane and some area increase brought by the fusion of LUVs to the freestanding membrane?
- The captions should contain details on compositions and concentrations, as the membrane composition (DOPC with ??% the fluorescent dye XXX), concentrations of sucrose, KCl, NaCl, Alexa probe and lipid of added LUVs, etc.
- The section "Discussion" is not a Discussion, but rather "Conclusion"
- In the methods, describe how encapsulation of Alexa was done, only via dilution? Give the lipid concentrations used. Change "SUVs solution" to "dispersion" or "suspension".
- Legend Fig. 1: Reread item iii)

Reviewer #3 (Remarks to the Author):

Kim and coworkers report on a modification of a previous protocol to generate 3DFLBS. In the previous publication (ACS Appl. Mater. Interfaces 2018, 10, 40401–40410) they used SU8 microwells applying an AC electric field. In the current manuscript, they additionally applied hydraulic pressure to improve the control over the shape and size of the 3D freestanding lipid bilayer structures (3DFLBS). Even though this

might be an interesting additional parameter to vary, the structures are still very similar to those of the previous publication.

Besides this, I have some problems with the membranes themselves. The observed structures and schematic drawings suggest different architectures on the surface. Fig. 1b suggests that the membranes are inside the wells and do not cover the top part of the SU-8. However, Fig. 1c shows that there is lipid material in between the 3DFLBS. Fig 1d even shows that there is a fluorescent layer (lipid material) in between the 3DFLBS connecting them. What is the structure on the surface? The same holds true for Fig. 2a and b. It remains unclear, which structure is formed at the edges of the microwells. In Fig. 2a v) there is no lipid membrane at the bottom?

In their previous study, the authors used α -hemolysin to demonstrate that single lipid bilayers were formed. Here, they used melittin, which is known to permeabilize the membrane (a carpet-like mechanism is proposed). In this case, permeabilization is very likely, even if a perfect lipid bilayer is not formed. α -hemolysin would be the better choice.

Concerning the fusion experiment (Figure 4e), it cannot be unambiguously concluded that full fusion has taken place as suggested in the schematic drawing. How can the authors rule out that the addition of the vesicles does not make the 3DFLBS leaky so that Alexa fluor 488 diffuses into the 3DFLBS? Is there another proof that full fusion occurs, and not docking/hemifusion of the SUVs? Side views should be shown in Fig. 4b and e.

In both manuscripts (2018 and the current one), the authors use DOPC. To make the method more universal, other lipids would need to be implemented. Is the procedure limited to DOPC or can other lipids or lipid mixtures be used?

Throughout the manuscript, I am missing the statistics. How often are the structures observed? What are the standard deviations of the curves shown in Fig. 4b and f?

Response to the reviewer 1's comments

The paper addresses an important and timely technological problem. The authors present an alternative electroformation protocol by implementing an electric field with hydraulic pressure to form giant-unilamellar vesicles in microwells. The method is interesting and it is important that we better understand the fusion and growth of vesicles, but the presented data are rather preliminary and do not address key issues in the field. The challenge is not so much to form vesicles from DOPC lipid but to form vesicles from physiologically relevant lipid mixtures with integral membrane proteins embedded in the lipid bilayer and to encapsulate components reproducibly and with high efficiency. The present paper does not address these issues.

[RESPONSE] We appreciate the reviewer's thoughtful evaluation of our work. We totally agree with the current key issues to solve in the field that the reviewer pointed out. For a long time, GUV fabrication has been well discovered using simple hydration, electroformation only or other assistance method as a solvent free manner for membrane protein stability. There is no doubt that floating GUV is an excellent tool. While GUVs, however, offer valuable opportunities in various fields including synthetic biology and life science, their inherent mobility poses challenges for accurately measuring ion flux or molecular transports between membranes via integral membrane proteins. These transports through membrane proteins are pivotal across diverse scientific disciplines, including life science, brain science, and molecular cell biology, as outlined in response to comment 6. Many researchers pursue solving those issues by using 2D free-standing lipid bilayer structures or aperture structures on the substrate. However, the 2D structure is not stable and fragile when exchanging solutions for preparing lipid mixtures with membrane proteins or measuring electrophysiological signals. This paper seeks to solve those issues and, at the same time, to expand to device applications using lipid structures with any shape of the lipid bilayer, i.e., GUV shape, oval shape, or flat lipid bilayer, creating stable 3D freestanding lipid bilayer (3DFLB) structures on solid substrates. **The novelty of this study is that it offers a solution to fabricate such a stable 3D lipid bilayer structure by controlling membrane fusion through manipulating pressure and electric field. 3DFLB will also offer electrophysiological measurements of such ion transports by integrating electrodes in it in the future.**

The use of DOPC lipid in our study is because it is the most commonly employed material for electroformation. Nonetheless, our proposed method effectively extends to various lipid materials and their mixtures. According to reviewer, we verify our approach with different lipid compositions, such as 18:0-18:1 PC, 16:0-18:1 PE, DOPC-DOPE mixture, and DOPC-DOPE-DOPS mixture through additional experiments. While this paper primarily focuses on demonstrating fusion control in physiological solutions through electroformation and hydraulic pressure, overcoming conventional methods, we acknowledge the reviewer's perspective and have revised the manuscript to provide additional details, as outlined in response to comment 2.

Finally, as one of the solutions for the reviewer's concern regarding the creation of lipid bilayers incorporating membrane proteins, we introduced a relatively straightforward approach using hydraulic pressure to fuse small unilamellar vesicles (SUVs) to 3DFLBs, as depicted in Figure 4d-f. This method verifies the potential recombination of SUVs or proteoliposomes containing various cell components into 3DFLBs, indicating its utility in constructing artificial cell membranes with diverse cell components, as outlined in response to comment 3.

In conclusion, the primary objective of this paper is to overcome the limitations of conventional electroformation in physiological solutions by regulating membrane fusion and fabricating solvent-free freestanding lipid bilayers. The three-dimensional freestanding lipid bilayers generated through our method exhibit remarkable stability and broad applicability. While the reviewer rightfully raises the issue of lipid mixtures with integral membrane proteins embedded in the lipid bilayer, we recognize that addressing this matter extends beyond the scope of our work in this paper and should be explored in future studies.

1. The statement in the introduction that "... electroformation is an almost sole and representative method to produce solvent-free lipid bilayer membrane ..." is not correct. There are various alternative methods such as gel-assisted swelling that offer an alternative but, like electroformation, has limitations.

[RESPONSE] Thank you for the reviewer's meticulous comments. Indeed, gel-assisted swelling, as mentioned by the reviewer, can be used to create solvent-free lipid bilayer membranes. However, it's important to note that this method is employed for the production of floating vesicles, such as GUVs (Giant Unilamellar Vesicles). The focus of our paper is on freestanding lipid bilayer membranes that allow the separation of two compartments (cis/trans chamber) across the lipid membrane, distinct from GUVs.

In a previous study by Jennifer Schultze *et al.* (ACS Omega 2019, 4, 5, 9393–9399), lipid rehydration onto

micropatterned hydrogel substrate was utilized to create anchored vesicles of relatively uniform size. However, this work did not explicitly address encapsulation (sealing), and to our knowledge, the penetration of molecules into the anchored GUV via the bottom hydrogel substrate makes it challenging to achieve a clear separation between the internal and external environments. Thus, it deviates from what we refer to as a freestanding lipid bilayer membrane.

Our intended message in the manuscript is that electroformation is one of the most unique and representative methods capable of producing solvent-free freestanding lipid bilayer membranes. To prevent any misinterpretation by the readers, we have revised the manuscript as follows.

[MODIFICATION]

“... rehydration of dried lipid stacks with electric fields, i.e., electroformation, is an almost sole and representative method to produce a solvent-free lipid bilayer membrane ...”

to

“... rehydration of dried lipid stacks with electric fields, i.e., electroformation, is a representative method to produce a solvent-free lipid bilayer membrane ...”

2. The advantage of the presented method over existing protocols is not very clear, except that one has an extra level of control by varying the pressure. Making vesicles by evaporating chloroform and hydrating DOPC lipids is relatively trivial and protocols are available to do this at physiological salt concentrations. It is more challenging to make vesicles from physiologically relevant amphiphile mixtures, including anionic and non-bilayer type of lipids and sterols, which are not tested in this paper. How does the method perform with other lipid mixtures.

[RESPONSE] Thank you for the reviewer's precious comment. This paper does not simply make conventional vesicles with dried lipid. Our assertion in this paper is that by adjusting the combination of electroformation and hydraulic pressure, fusion can be controlled even in a physiological solution to fabricate the desired freestanding lipid bilayer membrane with proper shape and size distribution anchored on the substrate. To demonstrate this, we explored various conditions of electric fields and hydraulic pressures using DOPC and confirmed the applicability of our proposed method. In order to clarify our proposed method based on the reviewer's comment, we have added results for various lipids and lipid mixtures in the manuscript. Non-bilayer lipid PE has been included in Figure 3j, and the mixture of DOPC and DOPE is depicted in Figure 3k. Additionally, Figure 3l illustrates the formation of 3DFLB using a mixture containing charged lipid DOPS. Added data demonstrates that our work can be applied to various lipid materials and their mixtures in the same manner.

[MODIFICATION] We added/modified some sentences and figures to the manuscript as below

Added sentences in the “Results” section

“In addition to DOPC, the synergy of microwell structures, an electric field, and hydraulic pressure enables the precise control of membrane fusion, leading to the fabrication of 3DFLBs. Various lipid compositions, including bilayer lipid (18:0-18:1 PC; Fig. 3i), non-bilayer lipid (16:0-18:1 PE; Fig. 3j), a mixture of bilayer and non-bilayer lipid (DOPC:DOPE=50:50; Fig. 3k), and even a mixture containing charged lipid (DOPC:DOPE:DOPS=50:45:5; Fig. 3l), have all successfully produced 3DFLBs with a high level of tight sealing using the same methodology.”

Modified sentence in the “Materials” section

“All lipids used, namely, 1,2-dioleoyl-sn-glycero-3-phosphocholine (DOPC), 1,2-dioleoyl-sn-glycero-3-phosphoethanolamine-*N*-(lissamine rhodamine B sulfonyl) (Rhod-PE), and 1,2-dioleoyl-sn-glycero-3-phosphoethanolamine-*N*-(7-nitro-2-1,3-benzoxadiazol-4-yl) (18:1 NBD-PE), were purchased from Avanti Polar Lipids, Inc. (USA).”

to

“All lipids used, namely, 1,2-dioleoyl-sn-glycero-3-phosphocholine (DOPC), 1-stearoyl-2-oleoyl-sn-glycero-3-phosphocholine(18:0-18:1 PC), 1,2-dioleoyl-sn-glycero-3-phosphoethanolamine (DOPE), 1-palmitoyl-2-oleoyl-sn-glycero-3-phosphoethanolamine (16:0-18:1 PE), 1,2-dioleoyl-sn-glycero-3-phospho-L-serine (DOPS), 1,2-dioleoyl-sn-glycero-3-phosphoethanolamine-*N*-(lissamine rhodamine B sulfonyl) (Rhod-PE), and 1,2-dioleoyl-sn-glycero-3-phosphoethanolamine-*N*-(7-nitro-2-1,3-benzoxadiazol-4-yl) (18:1 NBD-PE), were purchased from Avanti Polar Lipids, Inc. (USA).”

Modified/added sentences in the “Patterning of lipids in the microwell array” section

“An appropriate amount of chloroform was added to a vial containing DOPC with 1 mol% Rhod-PE for fluorescence observation to obtain 17 mM lipid solutions.”

to

“An appropriate amount of chloroform was added to a vial containing lipid or lipid mixture with 1 mol% Rhod-PE for fluorescence observation to obtain 17 mM lipid solutions. The lipid we used is DOPC, 18:0-18:1 PC, or 16:0-18:1 PE. The lipid mixture we used consisted of 50 mol% DOPC and 50 mol% DOPE or 50 mol% DOPC, 45 mol% DOPE, and 5 mol% DOPS.”

Added data in Figure 3i to l

Added sentences in legend of Figure 3

“(i-l) Representative cross-sectional confocal fluorescence microscopy images and corresponding tight-sealing tests of 3DFLBs generated in KCl solution with different lipid compositions as (i) 18:0-18:1 PC, (j) 16:0-18:1 PE, (k) a mixture of DOPC (50 mol%) and DOPE (50 mol%), and (l) a mixture of DOPC (50 mol%), DOPE (45 mol%), and DOPS (5 mol%). The 3DFLBs with various lipids were generated under the same conditions as 10 kHz frequency and a hydraulic pressure of 1.52 kPa. Scale bar: 5 μ m.”

3. Even more challenging is to form micrometer-size vesicles with genuine membrane proteins embedded in the bilayer. This requires forming vesicles from solvent-free membranes in which proteins have been incorporated. Again, protocols are available for making membrane protein-containing giant-unilamellar vesicles, but the efficiency is low and the variability is high. Any new method that addresses these problems is very welcome. The paper would gain enormous value if the functioning and quantitative analysis of real membrane receptors, channels and enzymes would have been tested. The autoinsertion of peptides and the pore formation by melittin (Figure 4) is not a real advance.

[RESPONSE] We appreciate the reviewer's perspective. Various protocols have been developed and reported for creating micrometer-sized vesicles containing membrane proteins, as mentioned by the reviewer, but these protocols often suffer from low efficiency and high variability. We agree that reconstituting a purified membrane protein into micrometer-sized vesicles is a challenging task, and numerous studies address this difficulty.

Approaching this issue from a bottom-up synthesis perspective, it is available to create membrane proteins embedded small vesicles (proteoliposomes) using methods tailored to each protein type. Subsequently, after undergoing filtration, concentration, and so on steps to enhance the low yield, these small vesicles (proteoliposomes) can be inserted into the target large membrane through methods such as fusion, resulting in the

creation of a micrometer-sized lipid bilayer containing membrane proteins.

Although various research on such methods has been published, one common issue is that membrane fusion rarely occurs in normal circumstances of lipid membranes. In our previous study, we addressed this by using charged lipids to increase the fusion rate and successfully reconstituted 5HT-3A (serotonin receptor) into the 3DFLBs, confirming protein activity. **In this paper, we have elevated the fusion rate between small unilamellar vesicles and 3DFLBs using hydraulic pressure, as shown in Figures 4d-f.** That is a simple and more universal method compared to our previous work via charged lipids. This demonstrates the potential for reassembling various cell components, including membrane proteins, into artificial cell membranes.

Regarding the reviewer's mention of melittin pore formation, it serves as a commonly known method to confirm the unilamellarity and biofunctionality of lipid membranes (Deng, N.-N., Yelleswarapu, M., & Huck, W. T. S. (2016). Monodisperse Uni- and Multicompartment Liposomes. *Journal of the American Chemical Society*, 138(24), 7584–7591). Figures 4a-c in our work illustrate this aspect of confirming the unilamellarity of the fabricated lipid membranes.

4. Why are 50 nm SUVs used as starting materials? Would the method work with larger vesicles (extrusion through 100, 200 or 400 nm polycarbonate filters), which are more commonly used in functional studies and allow more protein to be incorporated? Is the high curvature of SUVs important for fusion of with the 3DFLBs? How many SUVs fuse? What is the experimental variation? Can quantitative data be presented?

[RESPONSE] In the preparation steps for generating 3DFLBs, we selectively coated lipids into SU-8 microwells with a lipid solution containing 50 nm SUVs, which were not for the functional study of lipid membranes. Utilizing the pinning phenomenon based on the contact angle, as described in our previous paper, we selectively infused the lipid solution containing 50 nm SUVs only into the microwells, followed by drying to form a uniform lipid stack within the microwell array. The purpose of choosing 50 nm SUVs is to achieve a uniform coating of lipids into the microwell array. Using larger SUVs created by extrusion through 100–400 nm size polycarbonate filters led to lower uniformity due to larger suspension vesicle sizes. Thus, we used the lipid solution consisting of DI water base containing suspended 50 nm SUVs that ensure high uniformity in selective coating into the microwell array. (see below selective coating images in microwells)

Figure 4d illustrates the controlled fusion of 100 nm SUVs and the fully generated 3DFLB by hydraulic pressure, which is mimetic of vesicular transport in an in-vitro manner. As mentioned in the response of comment 3, research on the reconstitution of cell components, such as membrane proteins, into artificial cell membranes through vesicle fusion has been extensively conducted and reported. Ref. 39 in the manuscript suggests that small curvature radius is advantageous for vesicle adhesion, and adhesion between SUVs and GUVs occurs more easily than between GUVs. However, even if adhesion between SUVs and GUVs occurs, fusion does not occur without the assistance of proteins like SNARE for the fusion of desired cell components into vesicles. We demonstrated the potential of our method for controlling the fusion of SUVs into artificial membranes by applying hydraulic pressure.

In addition, we validated our approach through control experiments with and without applied hydraulic pressure. However, quantitative assessment of SUV fusion is challenging in real-time due to the small size of SUVs (they

can't observe any microscopy). Therefore, we evaluated the fluorescence intensity of Alexa Fluor 488 influx into the 3DFLB through SUV fusion. By this evaluation, we can infer SUV fusion rate by an increase in fluorescence intensity, but it is inaccurate.

5. The comparison of the electric field-hydraulic pressure method to form vesicles with SNARE-based fusion is far-fetched and not very relevant for the here presented protocol.

[RESPONSE] The key to our work lies in the control of membrane fusion. *In-vivo*, membrane fusion is facilitated by various fusion proteins or peptides, with SNARE proteins being a prominent example. However, pure lipid membrane fusion, as explained in the manuscript, typically requires 10 MPa of pressure for hemifusion (ref. 22 in the manuscript), and 100 MPa for complete fusion (ref. 23 in the manuscript), as reported in previous studies. In contrast to these precedents, our work, involving the combination of microwells, hydraulic pressure, and an electric field, promotes membrane fusion at remarkably lower pressures. This is akin to the role of SNARE proteins in *in-vivo* cell fusion. When SNARE proteins induce membrane fusion, an additional tension of 3.4-5.0 mN/m is exerted (ref. 27 in the manuscript). By calculating, in our work, applied hydraulic pressure generates tension on the lipid bilayer membrane ranging from 0.83-6.29 mN/m (Fig. 2c).

In summary, the membrane fusion control in our work, with a tension increase similar to the action of SNARE proteins, is highly effective. This comparison is meaningful, as it demonstrates that our method, with a tension increase comparable to SNARE-based fusion, is much more efficient than conventional methods requiring 50 mN/m for fusion control (ref. 28 in the manuscript). This suggests that our approach is highly effective in controlling membrane fusion, similar to the efficiency seen with *in-vivo* SNARE-mediated fusion.

6. The authors mention the formation of tightly-sealed vesicle as a major advantage of the new method over conventional electroformation. However, electroformation also yields well-sealed vesicles that can be used to form and maintain electrochemical ion gradients. What is different here?

[RESPONSE] Unlike the floating GUV mentioned by the reviewer, 3D FLB fixed to the substrate is already known to be an important issue that must be resolved as it is difficult to maintain sealing due to contact with the substrate. As reiterated in the responses, our focus in the paper is on freestanding lipid bilayers immobilized on a substrate that allows the separation of two compartments (cis/trans chamber), not floating GUVs. Rehydration of dried lipid stack is a representative method for creating solvent-free freestanding lipid bilayers. However, lipid membranes generated by rehydration are leaky due to the nanometer gap between the substrate and the lipid membrane or between lipid membranes, allowing molecules to easily pass through, and they do not effectively encapsulate or separate two compartments by a lipid bilayer.

In our previous research, we addressed this issue by adjusting microwells and electric fields in sucrose solution, achieving encapsulation (separation of two compartments by 3DFLBs). However, in solutions like physiological ones containing ions, interference from ions hinders effective membrane fusion, making it challenging to achieve tight sealing of 3DFLBs. In this work, we demonstrated the method to control membrane fusion even in physiological solutions, creating the desired 3DFLBs and achieving tight sealing by hydraulic pressure. Furthermore, we addressed the limitations of the conventional electroformation method for creating GUVs in physiological solutions, overcoming size variation and uncontrollability issues. This allowed us to precisely control and produce monodisperse-sized 3DFLBs.

Minor points:

- The 3DFLBS is not explained and not common terminology to present vesicles. Are 3DFLBSs different from giant-unilamellar vesicles?

[RESPONSE] Thank you for your careful comments. "Freestanding lipid bilayer" is a term already in use. From our previous works, we named our lipid bilayer structures as 3DFLBs (3-dimensional freestanding lipid bilayers) to emphasize that it was fabricated in three dimensions. As explained in the previous response, this is different from floating GUVs. Ours are fixed to the substrate and separated into two compartments by a lipid bilayer.

- What is DPBS solution?

[RESPONSE] PBS is a commonly used buffer with a simple composition. On the other hand, DPBS, which stands for Dulbecco's Phosphate-Buffered Saline, is more suitable for cell culture. It includes KCl to reduce cell toxicity and eliminates Ca^{2+} to prevent cell adhesion mediated by Ca^{2+} .

Response to the reviewer 2's comments

In this contribution the authors describe the swelling of freestanding bilayers through fusion controlled by electric field and hydraulic pressure in different media/conditions. A lipid film is formed on a chip with an array of 8 μ m-diameter microwells and application of an AC field and hydraulic pressure causes the swelling of freestanding bilayers, whose size/shape depend on field and pressure parameters, as well as ionic concentration. In the absence of hydraulic pressure, swelling readily occurs giving rise to large and multilayered structures. But, under certain conditions of AC field, hydraulic pressure and medium composition, a well-defined and tightly sealed 3D freestanding bilayer is created on top of every microwell. Apart from defining the conditions for a controlled fusion-mediated swell, some bioinspired applications are shown, such as interaction with a lytic peptide, fusion-mediated vesicular transport and long-term stability.

The work is nicely done and provides a high-quality and comprehensive description of the swelling of freestanding bilayers via fusion of adjacent lamellae as a function of electric field frequency, hydraulic pressure and medium composition. The results are robust and the method gives rise to very well-defined structures modulated by the microchip array and the above-mentioned parameters. My main comment/concern regards the main purpose of the work and, as a consequence, the interest/applicability to the broad community of this very high standard journal. It was not clear to me whether the main purpose of the work was to describe a well-controlled way to provide 3D freestanding bilayers so that others could reproduce the method and apply to their own interest, or whether the main goal was to study the fusion process under the conditions explored. Either way, the work and its applicability/relevance remain restricted to the use of a very specific combination of setups, such as the microwell chip, the hydraulic pressure control and the coupled observation system. The method for obtaining the freestanding bilayers on the microwell platform was already published by the same group (ref. 18), and the current work adds the hydraulic pressure which enhances fusion of the swelled structures. On the other hand, the relevance of the used parameters to the fusion process are very specific for the current system, and might not be general enough to apply in other situations.

[RESPONSE] Thank you for the thoughtful comments from the reviewer. Our research focuses on the generation of solvent-free freestanding lipid bilayers in physiological conditions containing ions, significantly enhancing their stability for diverse applications. As described in the manuscript, the key is to control membrane fusion during the rehydration process of dried lipid stacks, and we introduced microwells, an electric field, and hydraulic pressure for precise control.

In our previous study (ref. 18 in the manuscript), we formed 3DFLBs using microwells and electric fields, but these were generated **in a sucrose solution**. Controlling the generation of 3DFLBs in physiological solutions, due to ion interference, was challenging. To overcome this, in this work, we introduced hydraulic pressure as an additional variable and successfully generated well-controlled freestanding lipid bilayers in an array form under physiological conditions. The importance of controlling membrane fusion in this process is crucial, and we demonstrated that the introduced hydraulic pressure can promote membrane fusion at much lower pressures than reported in other studies (ref. 22, 23, and 27 in the manuscript).

We agree our approaches look complex. However, our proposed method allows anyone in the community to easily produce well-defined freestanding lipid bilayers with adjustable shapes and sizes, achieving complete separation of internal and external compartments through robust sealing. Moreover, the stability of our approach surpasses that of conventional freestanding lipid bilayers (i.e. BLMs, DIBs), making it a meaningful platform not only for experiments on cell membrane proteins or artificial cells but also for industrial applications.

Addressing the reviewer's concerns, we empathize with them, and to demonstrate the universality of our proposed method, we have included additional results for various lipids and lipid mixtures in the manuscript. Non-bilayer lipid PE is presented in Figure 3j, and the mixture of DOPC and DOPE is illustrated in Figure 3k. Additionally, Figure 3l showcases the formation of 3DFLB using a mixture containing the charged lipid DOPS. This additional data confirms that our approach can be applied uniformly to various lipid materials and their mixtures.

[MODIFICATION] We added/modified some sentences and figures to the manuscript as below

Added sentences in the "Results" section

"In addition to DOPC, the synergy of microwell structures, an electric field, and hydraulic pressure enables the precise control of membrane fusion, leading to the fabrication of 3DFLBs. Various lipid compositions, including bilayer lipid (18:0-18:1 PC; Fig. 3i), non-bilayer lipid (16:0-18:1 PE; Fig. 3j), a mixture of bilayer and non-bilayer

lipid (DOPC:DOPE=50:50; Fig. 3k), and even a mixture containing charged lipid (DOPC:DOPE:DOP=50:45:5; Fig. 3l), have all successfully produced 3DFLBs with a high level of tight sealing using the same methodology.”

Modified sentence in the “Materials” section

“All lipids used, namely, 1,2-dioleoyl-sn-glycero-3-phosphocholine (DOPC), 1,2-dioleoyl-sn-glycero-3-phosphoethanolamine-*N*-(lissamine rhodamine B sulfonyl) (Rhod-PE), and 1,2-dioleoyl-sn-glycero-3-phosphoethanolamine-*N*-(7-nitro-2-1,3-benzoxadiazol-4-yl) (18:1 NBD-PE), were purchased from Avanti Polar Lipids, Inc. (USA).”

to

“All lipids used, namely, 1,2-dioleoyl-sn-glycero-3-phosphocholine (DOPC), 1-stearoyl-2-oleoyl-sn-glycero-3-phosphocholine(18:0-18:1 PC), 1,2-dioleoyl-sn-glycero-3-phosphoethanolamine (DOPE), 1-palmitoyl-2-oleoyl-sn-glycero-3-phosphoethanolamine (16:0-18:1 PE), 1,2-dioleoyl-sn-glycero-3-phospho-L-serine (DOPS), 1,2-dioleoyl-sn-glycero-3-phosphoethanolamine-*N*-(lissamine rhodamine B sulfonyl) (Rhod-PE), and 1,2-dioleoyl-sn-glycero-3-phosphoethanolamine-*N*-(7-nitro-2-1,3-benzoxadiazol-4-yl) (18:1 NBD-PE), were purchased from Avanti Polar Lipids, Inc. (USA).”

Modified/added sentences in the “Patterning of lipids in the microwell array” section

“An appropriate amount of chloroform was added to a vial containing DOPC with 1 mol% Rhod-PE for fluorescence observation to obtain 17 mM lipid solutions.”

to

“An appropriate amount of chloroform was added to a vial containing lipid or lipid mixture with 1 mol% Rhod-PE for fluorescence observation to obtain 17 mM lipid solutions. The lipid we used is DOPC, 18:0-18:1 PC, or 16:0-18:1 PE. The lipid mixture we used consisted of 50 mol% DOPC and 50 mol% DOPE or 50 mol% DOPC, 45 mol% DOPE, and 5 mol% DOPS.”

Added data in Figure 3i to l

Added sentences in legend of Figure 3

“(i-l) Representative cross-sectional confocal fluorescence microscopy images and corresponding tight-sealing tests of 3DFLBs generated in KCl solution with different lipid compositions as (i) 18:0-18:1 PC, (j) 16:0-18:1 PE, (k) a mixture of DOPC (50 mol%) and DOPE (50 mol%), and (l) a mixture of DOPC (50 mol%), DOPE (45 mol%), and DOPS (5 mol%). The 3DFLBs with various lipids were generated under the same conditions as 10 kHz frequency and a hydraulic pressure of 1.52 kPa. Scale bar: 5 μm.”

Specific issues:

- Add the section “Introduction” after the abstract.

[RESPONSE] Our manuscript is structured in accordance with the guidelines of Nature Communications,

which does not include a heading for "Introduction".

- The abbreviation 3DFLBS is only explained in the caption of Figure 1, but the whole name should appear the first time the abbreviation is used in the Introduction. The choice of the abbreviation might not be the best as well, since it is too long and not intuitive of its meaning. In previous publications the authors preferred 3D FLBs and 3DFLB.

[RESPONSE] Thank you for the meticulous observation. It was an oversight not to include the full name of 3DFLBS in the main text. We agree with the reviewer's suggestion, and as per your recommendation, we have revised the abbreviation to 3D freestanding lipid bilayer (3DFLB), as used in our previous publications, for brevity and clarity.

[MODIFICATION] We added the full name of 3DFLBs in the manuscript, and all the abbreviations of 3DFLBS were revised as 3DFLB.

- Small liposomes are always called SUVs, irrespective of their size. In a section they are said to have 100 nm and in another 200 nm. Yet in the methods a 50 nm-pore membrane is used in the extrusion process. Conventionally, 100nm and 200nm liposomes are called LUVs, and SUVs usually have 30-50nm diameter. Additionally, even when using a 50nm-pore membrane for extrusion, usually vesicles closer to 100nm diameter are obtained. SUVs are usually formed using a tip sonicator. So I guess all liposomes described are rather LUVs.

[RESPONSE] Thank you for the reviewer's comment. The mention of "200 nm" in "Mimetics of vesicular transport" in the "Method" section was a typo, and it should be 100 nm pore membrane (We corrected it to "100 nm").

We used a 50 nm-pore membrane for extruding to create small unilamellar vesicles (SUVs) for selective coating of lipids into microwells. To mimic vesicular transport, we utilized a 100 nm-pore membrane for extruding. Typically, the distinction between SUVs and LUVs is made at 100 nm, with SUVs having a diameter below 100 nm and LUVs ranging from 100 to 1000 nm.

We measured the diameter of the unilamellar vesicles produced after extrusion with each size of the pore membrane. After extruding with the 50 nm-pore membrane, the diameter of the SUVs was 49.39 nm. Following extruding with the 100 nm-pore membrane, the diameter of the SUVs was 94.43 nm. All vesicles produced through our extrusion process fall below 100 nm in size, resulted in that we can classify them as SUVs.

[MODIFICATION]

"... followed by extrusion through a 200 nm polycarbonate membrane filter..."

to

“... followed by extrusion through a 100 nm polycarbonate membrane filter...”

- In the experiment mimicking vesicular transfer via fusion, specify the types of fluorophores in the LUVs: The lipid membrane is labeled with 1 mol% NBD-PE and the LUVs encapsulate the aqueous soluble probe Alexa 488. The phrase “SUVs containing Alexa Fluor 488 and NBD-PE in the membrane” does not completely specify the system.

[RESPONSE] Thank you for the reviewer's feedback. To prevent any misunderstandings and enhance clarity, we have revised the sentence in the manuscript (Legend of Figure 4d).

[MODIFICATION]

“SUVs containing Alexa Fluor 488 and NBD-PE in the membrane”

to

“NBD-PE (1 mol%) labeled SUVs containing Alexa Fluor 488 (5 μ M)”

- Related to the previous comment, why don't we see clear evidence of lipid mixing in Fig. 4? Wouldn't we expect to have some green fluorescence on the membrane and some area increase brought by the fusion of LUVs to the freestanding membrane?

[RESPONSE] Thank you for the reviewer's feedback. The size of SUVs is approximately 100 nm in diameter, with an area of 0.03 μ m². In contrast, the 3DFLBs in experiments shown in Figures 4d-f have a semi-spherical shape with a diameter of 14 μ m and an area of 560.54 μ m². The area of 3DFLBs is approximately 17,842 times larger than that of SUVs. To observe an increase in area corresponding to the size of 3DFLBs, at least a 10% increase is expected, requiring the fusion of approximately 1,700 or more SUVs. Our experiments confirmed the fusion of SUVs to 3DFLBs induced by hydraulic pressure, and the increase in green fluorescence due to NBD-PE in 3DFLBs was at the level of the noise. Although an increase in green fluorescence of 3DFLBs could be expected with a sufficient fusion time for SUVs, our study aims to conceptually demonstrate the mimicry of vesicular transport.

- The captions should contain details on compositions and concentrations, as the membrane composition (DOPC with ???% the fluorescent dye XXX), concentrations of sucrose, KCl, NaCl, Alexa probe and lipid of added LUVs, etc.

[RESPONSE] We thank the reviewer for the detailed comment. To provide more information, we modified/added some sentences in figure legends.

[MODIFICATION] We added some sentences in each figure legend to provide materials and their concentrations as below

Added sentence in Fig. 1

“Lipid composition: DOPC with 1 mol% Rhod-PE.”

Added sentence in Fig. 2

“Lipid composition: DOPC with 1 mol% Rhod-PE.”

Modified and Added some sentences in Fig. 3

“(b) 10 mM sucrose solution and (c) KCl solution (140 mM KCl, 10 mM HEPES; physiological ionic condition: 300 mOsm/L).”

“(g) NaCl solution (140 mM NaCl, 10 mM HEPES) and (h) 150 mM DPBS solution...”

Added sentences in Fig. 4

“Lipid composition: DOPC with 1 mol% Rhod-PE.”

“Lipid composition: (3DFLB) DOPC with 1 mol% Rhod-PE, (SUV) DOPC with 1 mol% NBD-PE containing 5 μ M Alexa Fluor 488.”

Added sentences in Fig. 5

“Lipid composition: DOPC with 1 mol% Rhod-PE.”

- The section “Discussion” is not a Discussion, but rather “Conclusion”

[RESPONSE] Our manuscript is structured in accordance with the guidelines of Nature Communications, which does not include a heading for "Conclusion".

- In the methods, describe how encapsulation of Alexa was done, only via dilution? Give the lipid concentrations used. Change “SUVs solution” to “dispersion” or “suspension”.

[RESPONSE] Dried lipid film can be swollen by Alexa Fluor 488-containing KCl solution to produce multilayered vesicles (MLVs) encapsulating Alexa Fluor 488. These MLVs are then extruded to create 100 nm SUVs containing Alexa Fluor 488. To enhance clarity in conveying this information, we have made modifications to several sentences in the "Methods" section. We agree with the reviewer's suggestion and have changed "SUV solution" to "SUV dispersion" for improved terminology.

[MODIFICATION]

“The dried lipid film was hydrated with 1 mL KCl solution (140 mM KCl, 10 mM HEPES) containing fluorescent dye (5 μ M Alexa Fluor 488) under gentle agitation, followed by extrusion through a 100 nm polycarbonate membrane filter...”

“After confirmation of the tight sealing of 3DFLBs using KCl solution (140 mM KCl, 10 mM HEPES) containing fluorescent dye (3 μ M Alexa Fluor 488), the SUV dispersion was infused into the 3DFLBs through a microchannel...”

- Legend Fig. 1: Reread item iii)

[RESPONSE] We appreciate the reviewer's observation. We have identified a grammatical error in the description of item iii) in Figure 1a and have made the following correction.

[MODIFICATION]

“iii) hemifusion stalk: sufficiently proximity of membranes causes destabilized boundary between the hydrophilic and hydrophobic portion of the bilayer, resulting in non-bilayer transition states are generated that culminate in the formation of stalk and fusion pore.”

to

“iii) hemifusion stalk: sufficiently proximity of membranes causes destabilized boundary between the hydrophilic and hydrophobic portion of the bilayer, resulting in non-bilayer transition states which are generated that culminate in the formation of stalk and fusion pore.”

Response to the reviewer 3's comments

Kim and coworkers report on a modification of a previous protocol to generate 3DFLBs. In the previous publication (ACS Appl. Mater. Interfaces 2018, 10, 40401–40410) they used SU8 microwells applying an AC electric field. In the current manuscript, they additionally applied hydraulic pressure to improve the control over the shape and size of the 3D freestanding lipid bilayer structures (3DFLBs). Even though this might be an interesting additional parameter to vary, the structures are still very similar to those of the previous publication.

[RESPONSE] Thank you for the reviewer's comment. It is important to clarify that our previous publication focused on the generation of 3DFLBs using sucrose solution and did not involve ion-containing physiological solutions. In physiological solutions, the limitations of conventional electroformation hindered the formation of tightly sealed 3DFLBs due to the challenges posed by ion strength, making controlled membrane fusion difficult. In contrast to our previous work, this work introduces hydraulic pressure as an additional variable to overcome the limitations of conventional electroformation. We successfully controlled membrane fusion, adjusting the size and shape of 3DFLBs and forming tightly sealed structures. While the final appearance of the 3DFLBs may seem similar, the significant achievement lies in the successful control of membrane fusion in physiological solutions, overcoming a major challenge in electroformation.

Besides this, I have some problems with the membranes themselves. The observed structures and schematic drawings suggest different architectures on the surface. Fig. 1b suggests that the membranes are inside the wells and do not cover the top part of the SU-8. However, Fig. 1c shows that there is lipid material in between the 3DFLBs. Fig. 1d even shows that there is a fluorescent layer (lipid material) in between the 3DFLBs connecting them. What is the structure on the surface? The same holds true for Fig. 2a and b. It remains unclear, which structure is formed at the edges of the microwells. In Fig. 2a v) there is no lipid membrane at the bottom?

[RESPONSE] We appreciate the reviewer's detailed comment. As reported in our previous studies, there is a self-spreading lipid bilayer on the surface of the SU-8 template (Ref. 18 in the manuscript). The schematic in Figure 1b is intended to emphasize the 3DFLBs and SU-8 microwell template, but we acknowledge the potential for misinterpretation and have revised the illustrations to indicate the presence of a self-spreading lipid bilayer on the surface of the SU-8 template.

In Figures 1c and d, we show 3D reconstructed confocal microscopy of the representative 3DFLBs to provide a 3D bird's-eye view. As in the previous studies, it is clear that the self-spreading lipid bilayer covers the surface of the SU-8 template. In Figure 1d, where the size of the 3DFLBs is very low and close to the height of the microwell edge. We added a cross-section view as a subset in Figures 1c and d to enhance clarity for readers.

In Figure 2a, we represented the self-spreading lipid bilayer on the SU-8 template surface in the schematic, which is also evident in the confocal microscopy image. To aid the reader's understanding, we added labels indicating the self-spreading lipid bilayer to the figure.

Regarding Figure 2a v), once the growth of 3DFLBs is complete, the lipid stacks inside the microwell are entirely consumed, forming the 3DFLBs. As observed in the confocal microscopy image, no fluorescence, other than autofluorescence, was detected at the bottom of the SU-8 microwell (Si substrate).

[MODIFICATION]

Modified Figure 1b

Modified Figure 1c and d

Modified Figure 2a

Modified Supplementary Figure S1

In their previous study, the authors used α -hemolysin to demonstrate that single lipid bilayers were formed. Here, they used melittin, which is known to permeabilize the membrane (a carpet-like mechanism is proposed). In this case, permeabilization is very likely, even if a perfect lipid bilayer is not formed. α -hemolysin would be the better

choice.

[RESPONSE] We appreciate the reviewer's comment. As reported in our previous study, we utilized α -hemolysin to assess the unilamellarity and biofunctionality of the lipid membrane. However, due to recent restrictions on the import of α -hemolysin in Korea, we were unable to obtain a supply for use in our experiments. Consequently, we opted for melittin to evaluate the unilamellarity and biofunctionality of the 3DFLBs we generated. Melittin, like α -hemolysin, has been commonly used to assess unilamellarity and biofunctionality (Deng, N.-N., Yelleswarapu, M., & Huck, W. T. S. (2016). Monodisperse Uni- and Multicompartment Liposomes. *Journal of the American Chemical Society*, 138(24), 7584–7591).

Concerning the fusion experiment (Figure 4e), it cannot be unambiguously concluded that full fusion has taken place as suggested in the schematic drawing. How can the authors rule out that the addition of the vesicles does not make the 3DFLBs leaky so that Alexa fluor 488 diffuses into the 3DFLBs? Is there another proof that full fusion occurs, and not docking/hemifusion of the SUVs? Side views should be shown in Fig. 4b and e.

[RESPONSE] We appreciate the reviewer's comment. In the experiment of mimicking vesicular transport, we verified SUV membrane fusion induced by hydraulic pressure by conducting a control experiment without applying hydraulic pressure. In Figure 4f, we plotted the increase in the internal fluorescence intensity of 3DFLBs as a result of SUV membrane fusion induced by hydraulic pressure. To avoid any misunderstandings and facilitate a better understanding, we have added confocal microscopy images of the control experiment in Supplementary Figure S14. When pressure is not applied, there is no increase in the internal fluorescence intensity of 3DFLBs. However, when pressure is applied, the internal fluorescence intensity of 3DFLBs increases due to the influx of Alexa Fluor 488 from the SUVs through membrane fusion. If complete fusion had not occurred and only docking or hemifusion had taken place, the Alexa Fluor 488 (contained in the SUVs) would not be observed inside the 3DFLBs. Additionally, we conducted a control experiment by infusing SUVs into the microchannel without applying hydraulic pressure, confirming that there was no increase in fluorescence intensity inside the 3DFLBs due to leakage.

By the way, to obtain side views of 3DFLBs, it is necessary to perform a reconstruction by scanning along the z-axis using confocal microscopy. For this, scanning along the z-axis process takes approximately 5 minutes. For continuous monitoring of the internal fluorescence intensity of 3DFLBs, that is not suitable.

[MODIFICATION] To enhance the understanding of the reader, we added and revised some sentences in the Supplementary information as below.

“... infused SUVs are sequentially fused into the 3DFLBs, enriching the green fluorescence intensity in sealed 3DFLBs (Fig. 4e).”

to

“... infused SUVs are sequentially fused into the 3DFLBs, enriching the green fluorescence intensity in sealed 3DFLBs (Fig. 4e; Supplementary Figure S14).”

Added figures in SI

In both manuscripts (2018 and the current one), the authors use DOPC. To make the method more universal, other lipids would need to be implemented. Is the procedure limited to DOPC or can other lipids or lipid mixtures be used?

[RESPONSE] We deeply appreciate the reviewer's perspective, and to showcase the universality of our proposed method, we have added results for various lipids and lipid mixtures in the manuscript. Non-bilayer lipid PE has been included in Figure 3j, and the mixture of DOPC and DOPE is depicted in Figure 3k. Additionally, Figure 3l illustrates the formation of 3DFLB using a mixture containing charged lipid DOPS. Added data demonstrates that our work can be applied to various lipid materials and their mixtures in the same manner.

[MODIFICATION] We added/modified some sentences and figures to the manuscript as below

Added sentences in the “Results” section

“In addition to DOPC, the synergy of microwell structures, an electric field, and hydraulic pressure enables the precise control of membrane fusion, leading to the fabrication of 3DFLBs. Various lipid compositions, including bilayer lipid (18:0-18:1 PC; Fig. 3i), non-bilayer lipid (16:0-18:1 PE; Fig. 3j), a mixture of bilayer and non-bilayer lipid (DOPC:DOPE=50:50; Fig. 3k), and even a mixture containing charged lipid (DOPC:DOPE:DOP=50:45:5; Fig. 3l), have all successfully produced 3DFLBs with a high level of tight sealing using the same methodology.”

Modified sentence in the “Materials” section

“All lipids used, namely, 1,2-dioleoyl-sn-glycero-3-phosphocholine (DOPC), 1,2-dioleoyl-sn-glycero-3-phosphoethanolamine-*N*-(lissamine rhodamine B sulfonyl) (Rhod-PE), and 1,2-dioleoyl-sn-glycero-3-phosphoethanolamine-*N*-(7-nitro-2-1,3-benzoxadiazol-4-yl) (18:1 NBD-PE), were purchased from Avanti Polar Lipids, Inc. (USA).”

to

“All lipids used, namely, 1,2-dioleoyl-sn-glycero-3-phosphocholine (DOPC), 1-stearoyl-2-oleoyl-sn-glycero-3-phosphocholine(18:0-18:1 PC), 1,2-dioleoyl-sn-glycero-3-phosphoethanolamine (DOPE), 1-palmitoyl-2-oleoyl-sn-glycero-3-phosphoethanolamine (16:0-18:1 PE), 1,2-dioleoyl-sn-glycero-3-phospho-L-serine (DOPS), 1,2-dioleoyl-sn-glycero-3-phosphoethanolamine-*N*-(lissamine rhodamine B sulfonyl) (Rhod-PE), and 1,2-dioleoyl-sn-glycero-3-phosphoethanolamine-*N*-(7-nitro-2-1,3-benzoxadiazol-4-yl) (18:1 NBD-PE), were purchased from Avanti Polar Lipids, Inc. (USA).”

Modified/added sentences in the “Patterning of lipids in the microwell array” section

“An appropriate amount of chloroform was added to a vial containing DOPC with 1 mol% Rhod-PE for

fluorescence observation to obtain 17 mM lipid solutions.”

to

“An appropriate amount of chloroform was added to a vial containing lipid or lipid mixture with 1 mol% Rhod-PE for fluorescence observation to obtain 17 mM lipid solutions. The lipid we used is DOPC, 18:0-18:1 PC, or 16:0-18:1 PE. The lipid mixture we used consisted of 50 mol% DOPC and 50 mol% DOPE or 50 mol% DOPC, 45 mol% DOPE, and 5 mol% DOPS.”

Added data in Figure 3i to l

Added sentences in legend of Figure 3

“(i-l) Representative cross-sectional confocal fluorescence microscopy images and corresponding tight-sealing tests of 3DFLBs generated in KCl solution with different lipid compositions as (i) 18:0-18:1 PC, (j) 16:0-18:1 PE, (k) a mixture of DOPC (50 mol%) and DOPE (50 mol%), and (l) a mixture of DOPC (50 mol%), DOPE (45 mol%), and DOPS (5 mol%). The 3DFLBs with various lipids were generated under the same conditions as 10 kHz frequency and a hydraulic pressure of 1.52 kPa. Scale bar: 5 μm.”

Throughout the manuscript, I am missing the statistics. How often are the structures observed? What are the standard deviations of the curves shown in Fig. 4b and f?

[RESPONSE] We have confirmed the reproducibility through repeated experiments. We have added this information to the figure captions, indicating the number of experiments conducted to derive the data. Figures 4c and f represent the average values observed from 6 individual 3DFLBs. The standard deviations are dedicated as error bars.

REVIEWER COMMENTS

Reviewer #1 (Remarks to the Author):

The three reviewers raised similar concerns about the novelty and applicability of the technology, but in my view the authors have addressed the comments reasonably well. Ambiguities in the methodology are better explained in the revised manuscript, and some additional data (different lipid compositions) are now presented. It remains to be seen if the method finds wide use, which depends on the possibilities to use proteoSUVs. However, the authors are well aware of the needs of the field and have the expertise to develop GUV technology further.

I have no further comments for the revised manuscript.

Reviewer #2 (Remarks to the Author):

The manuscript improved, but the issue about the interest/applicability to the broad community of this journal persists. The authors added new lipid compositions as requested by the reviewers, but actually only showed that it works for PE lipids as well. The percentage of charged lipid used is very low (5%) and do not pose a challenge. To validate the growth of a charged membrane, 30-50mol% would be required. And to show the broad applicability of the method, addition of cholesterol (~30mol%) to the lipid mixture would also be important. Yet, the novelty in respect to the previous publication is not substantial.

Small issues:

- The abbreviation 3DFLBS still appears in the Supporting Information.
- Use SUVs to describe the vesicles that passed through the 50-nm pores and are used to coat the microarray surface and LUVs for the vesicles that were prepared through the 100-nm pores and used in the vesicular transport / fusion experiments. The nomenclature is not sharp at below and above 100nm, but rather approximate values. So vesicles around 100nm should be called LUVs (which includes vesicle with 94nm diameter), and only those significantly smaller should be called SUVs.
- To be consistent with DOPC and DOPS, use SOPC for 18:0-18:1 PC and POPE for 16:0-18:1 PE. And why do the authors chose to prepare SOPC, then POPE and for the 1:1 mixture used DOPC:DOPE? What's the explanation?

- line 203: DOPC:DOPE:DOP=50:45:5 add the “S” of DOPS. Also, as mentioned above, 5mol% DOPS is not enough to characterize a “charged” system. Did the authors tried to grow compositions with higher charge, like 30-50%, and where not successful?
- There are still mentions to “SUV solution” (lines 321,322). Change to “dispersion”.
- What is the meaning of “Reconstitution” of melittin on the 3DFLBs? This term is generally applied for incorporation of a membrane protein into a bilayer structure for instance, and not to the lytic action of a membrane-active peptide such as melittin.
- line 391: “To observe this fluorescence effect in vesicles other than the 3DFLBs”, what fluorescence effect is it referring to, since this is the first sentence in the section?
- Figure S1: Change “Lipid solution” for “SUVs” or “SUVs dispersion”. Lipid solution gives the impression of lipid dissolved in chloroform. On the second snapshot, why are the SUVs of such different sizes, as they show a quite narrow size distribution? Explain in the caption what “DI” water mean.
- According to the journal guidelines “The main text of an Article should begin with a section headed Introduction of referenced text that expands on the background of the work (some overlap with the abstract is acceptable), followed by sections headed Results, Discussion (if appropriate) and Methods (if appropriate).” So please add the section “Introduction”.

Reviewer #3 (Remarks to the Author):

I appreciate the authors' efforts in elucidating the obtained membrane structures in more detail, which significantly helped with the readability of the manuscript.

However, my main concern is the fact that the resulting membrane structures are very similar to those previously reported by the group and this concern questions the significance and novelty of this work. In comparison to their prior research, where they operated at a hydraulic pressure of 0 kPa in sucrose, the current study explored again the influence of AC frequencies but this time in conjunction with hydraulic pressure, which enabled them to use not only sucrose but salt. This expansion has led to the discovery of a regime enabling the production of tightly sealed 3DFLBs in the presence of salt, offering the potential replacement of sucrose – a noteworthy advancement for protein reconstitution.

While acknowledging this progress, it's important to recognize that this achievement requires the independent control of an additional parameter, necessitating a more intricate setup. The membrane structures unveiled may pave the way for future experiments, yet their specific applications have not been demonstrated in this study, leaving the practical gain in knowledge somewhat moderate. A clear advancement of the current membrane structures would be the reconstitution of a transmembrane protein, which is known to be still a challenge for the community working with large vesicle structures such as GUVs. The authors used only melittin, a soluble peptide, which was a replacement for hemolysin (as they state in their response) and that they applied to show the unilamellarity of the 3DFLBs.

Response to the reviewer 1's comments

The three reviewers raised similar concerns about the novelty and applicability of the technology, but in my view the authors have addressed the comments reasonably well. Ambiguities in the methodology are better explained in the revised manuscript, and some additional data (different lipid compositions) are now presented. It remains to be seen if the method finds wide use, which depends on the possibilities to use proteoSUVs. However, the authors are well aware of the needs of the field and have the expertise to develop GUV technology further.

I have no further comments for the revised manuscript.

[RESPONSE] Thank you very for taking time to review the revised manuscript. Owing to valuable comments, we are very happy to improve the manuscript that the ambiguities in the methodology have been clarified and additional data on different lipid compositions have been modified, thus enriching the comprehensiveness of the study.

Once again, we extend our gratitude for your valuable feedback and for indicating that you have no further comments on the revised manuscript.

Response to the reviewer 2's comments

The manuscript improved, but the issue about the interest/applicability to the broad community of this journal persists. The authors added new lipid compositions as requested by the reviewers, but actually only showed that it works for PE lipids as well. The percentage of charged lipid used is very low (5%) and do not pose a challenge. To validate the growth of a charged membrane, 30-50mol% would be required. And to show the broad applicability of the method, addition of cholesterol (~30mol%) to the lipid mixture would also be important. Yet, the novelty in respect to the previous publication is not substantial.

[RESPONSE] We sincerely appreciate the reviewer's insightful comments. Electroformation, as a method for producing solvent-free GUV-sized lipid bilayers, encounters limitations when physiological solutions (high ion concentrations) and elevated proportions of charged lipids (as mentioned by the reviewer, typically ranging from 30-50 mol%) are introduced into the lipid mixture. This is primarily due to ion interference and repulsion among the charged lipids, which adversely affect the electroformation process.

To our knowledge, the highest concentration of DOPS (50 mol%) used in electroformation to fabricate GUVs was reported by Dahmani et al [1]. **However, it is important to note that their study was conducted using a 150 mM sucrose solution, not a physiological solution containing ions, and did not control GUV size.** Additionally, the yield and reproducibility of GUV fabrication with highly concentrated DOPS were not discussed. In a previous report by Rodriguez et al., they found that **when the proportion of DOPS reaches 20%, giant vesicles can no longer be formed by electroformation** [2]. They suggested that this limit could potentially be overcome with the gentle hydration method.

Generally, electroformation with lipid mixtures containing charged lipids, such as DOPS, is predominantly performed in sucrose solutions rather than physiological conditions, and typically utilizes lipid mixtures containing 20 mol% or less DOPS [3-6]. In the field of fabricating solvent-free lipid bilayers via electroformation, the requirement for the extra amplitude of AC electric fields and higher concentrations of deposited lipid mixtures to fabricate GUVs with high concentrations of DOPS (more than 20 mol%) presents unusual conditions.

Fortunately, since the last revision, **we have successfully increased the proportion of DOPS from 5 mol% to 30 mol%, resulting in the successful generation and tight sealing of 3DFLBs. Although the 30 mol% DOPS contained lipid mixture is a tough condition in electroformation with physiological solutions, not sucrose solutions, it can be achieved under the control of microwell and hydraulic pressure. This remarkable achievement by using pressure-assisted electroformation in the microwell significantly broadens the scope of conventional electroformation, representing a substantial contribution to the field.** Consequently, we have replaced Figure 3I, which previously depicted the results of the lipid mixture containing 5 mol% DOPS, with the results of the lipid mixture containing 30 mol% DOPS.

Regarding the broad applicability of the method, we fully agree with the reviewer's suggestion to include ~30 mol% cholesterol. We are currently conducting exciting research using lipid mixtures containing cholesterol and charged lipids to generate 3DFLBs. A glimpse of this ongoing research indicates that we can successfully control the generation and tight sealing of 3DFLBs using lipid mixtures containing 30 mol% cholesterol, as illustrated in the figure below. To visualize cholesterol distribution, we used green fluorescent-tagged cholesterol. As depicted below the figure, we have an interesting point that cholesterol tends to segregate in specific areas, around the inlet of microwell, which correlates with lipid bilayer curvature [7]. We hypothesize that **regulating cholesterol segregation requires introducing a new variable: temperature.** We believe that forming 5-30 μm -sized lipid bilayers using various lipid compositions and sterol contents holds significant implications in this field. However, **achieving this requires not only controlling electric fields and pressure but also temperature**, which extends beyond the scope of this manuscript. We aspire to provide fresh insights into forming artificial cell membranes through diverse lipid compositions and sterol content and will address this in future research endeavors.

References

- [1] DOPS <50 mol%, 150 mM sucrose: Dahmani, Ismail, Kai Ludwig, and Salvatore Chiantia. "Influenza A matrix protein M1 induces lipid membrane deformation via protein multimerization." *Bioscience reports* 39.8 (2019): BSR20191024.
- [2] DOPS <40 mol%, 330 mOsm sucrose: Rodriguez, Nicolas, Frédéric Pincet, and Sophie Cribier. "Giant vesicles formed by gentle hydration and electroformation: a comparison by fluorescence microscopy." *Colloids and Surfaces B: Biointerfaces* 42.2 (2005): 125-130.
- [3] DOPS 15 mol%, 1 mg/mL dextran in de-ionized water: Kusters, Ilja, Antoine M. van Oijen, and Arnold JM Driessen. "Membrane-on-a-chip: microstructured silicon/silicon-dioxide chips for high-throughput screening of membrane transport and viral membrane fusion." *ACS nano* 8.4 (2014): 3380-3392.
- [4] DOPS 20 mol%, 200 mM sucrose: Witkowska, Agata, Lukasz Jablonski, and Reinhard Jahn. "A convenient protocol for generating giant unilamellar vesicles containing SNARE proteins using electroformation." *Scientific reports* 8.1 (2018): 9422.
- [5] DOPS 20 mol%, 250 mM sucrose: Nikolaus, Jörg, et al. "Direct visualization of large and protein-free hemifusion diaphragms." *Biophysical journal* 98.7 (2010): 1192-1199.
- [6] DOPS 20 mol%, 300 mM glycerol: Estes, Daniel J., and Michael Mayer. "Giant liposomes in physiological buffer using electroformation in a flow chamber." *Biochimica et Biophysica Acta (BBA)-Biomembranes* 1712.2 (2005): 152-160.
- [7] Cholesterol cluster: Ryu, Yong-Sang, et al. "Reconstituting ring-rafts in budding-mimicking topography of model membranes." *Nature communications* 5.1 (2014): 4507.

[MODIFICATION] We modified some sentences and figures to the manuscript as below

Modified sentence in the "Results" section

"Various lipid compositions, including bilayer lipid (18:0-18:1 PC; Fig. 3i), non-bilayer lipid (16:0-18:1 PE; Fig. 3j), a mixture of bilayer and non-bilayer lipid (DOPC:DOPE=50:50; Fig. 3k), and even a mixture containing charged lipid (DOPC:DOPE:DOP=50:45:5; Fig. 3l), have all successfully produced 3DFLBs with a high level of tight sealing using the same methodology."

to

"Various lipid compositions, including bilayer lipid (SOPC; Fig. 3i), non-bilayer lipid (POPE; Fig. 3j), a mixture of bilayer and non-bilayer lipid (DOPC:DOPE=50:50; Fig. 3k), and even a mixture containing charged lipid (DOPC:DOPE:DOPS=50:20:30; Fig. 3l), have all successfully produced 3DFLBs with a high level of tight sealing using the same methodology."

Modified sentence in the "Method" section

"The lipid mixture we used consisted of 50 mol% DOPC and 50 mol% DOPE or 50 mol% DOPC, 45 mol% DOPE, and 5 mol% DOPS."

to

“The lipid mixture we used consisted of 50 mol% DOPC and 50 mol% DOPE or 50 mol% DOPC, 20 mol% DOPE, and 30 mol% DOPS.”

Modified data in Figure 3I

Modified sentence in legend of Figure 3

“...and (l) a mixture of DOPC (50 mol%), DOPE (45 mol%), and DOPS (5 mol%).”

to

“... and (l) a mixture of DOPC (50 mol%), DOPE (20 mol%), and DOPS (30 mol%).”

Small issues:

- The abbreviation 3DFLBS still appears in the Supporting Information.

[RESPONSE] Thank you for your thorough observation, and we sincerely apologize for the oversight. We have revised the Supporting Information to ensure consistency, replacing all abbreviations of "3DFLBS" with "3DFLB" accordingly.

- Use SUVs to describe the vesicles that passed through the 50-nm pores and are used to coat the microarray surface and LUVs for the vesicles that were prepared through the 100-nm pores and used in the vesicular transport / fusion experiments. The nomenclature is not sharp at below and above 100nm, but rather approximate values. So vesicles around 100nm should be called LUVs (which includes vesicle with 94nm diameter), and only those significantly smaller should be called SUVs.

[RESPONSE] Thank you for your feedback. In many papers and review articles in this field, unilamellar vesicles with sizes below 100 nm are commonly referred to as SUVs, while those between 100 and 1000 nm are termed LUVs [1-10]. Following the widely accepted definitions in this field, we have labeled both the vesicles extruded through 50 nm and 100 nm pores (measuring 43.39 nm and 94.43 nm, respectively) as SUVs in the manuscript, as they fall within the size range below 100 nm. We kindly ask you to consider the following references for further clarification.

References

[1] Li, Hewen, Tao Zhao, and Zhihua Sun. "Analytical techniques and methods for study of drug-lipid membrane interactions." *Reviews in analytical chemistry* 37.1 (2018): 20170012.

- [2] van Swaay, Dirk, and Andrew DeMello. "Microfluidic methods for forming liposomes." *Lab on a Chip* 13.5 (2013): 752-767.
- [3] Ortega, Vanessa, Selma Giorgio, and Eneida de Paula. "Liposomal formulations in the pharmacological treatment of leishmaniasis: a review." *Journal of liposome research* 27.3 (2017): 234-248.
- [4] Voskuhl, Jens, and Bart Jan Ravoo. "Molecular recognition of bilayer vesicles." *Chemical Society Reviews* 38.2 (2009): 495-505.
- [5] Garrós, Núria, et al. "Baricitinib Liposomes as a New Approach for the Treatment of Sjögren's Syndrome." *Pharmaceutics* 14.9 (2022): 1895.
- [6] Mozafari, M. R., E. Mazaheri, and K. Dormiani. "Simple equations pertaining to the particle number and surface area of metallic, polymeric, lipidic and vesicular nanocarriers." *Scientia pharmaceutica* 89.2 (2021): 15.
- [7] Sapala, Appa Rao, Sameer Dhawan, and V. Haridas. "Vesicles: self-assembly beyond biological lipids." *RSC advances* 7.43 (2017): 26608-26624.
- [8] Mahajan, Srushti, et al. "Vesicular Nanomaterials: Types and Therapeutic Uses." *Nanomaterial-Based Drug Delivery Systems: Therapeutic and Theranostic Applications*. Cham: Springer International Publishing, 2023. 99-145.
- [9] Luiz, Hugo, Jacinta Oliveira Pinho, and Maria Manuela Gaspar. "Advancing Medicine with Lipid-Based Nanosystems—The Successful Case of Liposomes." *Biomedicines* 11.2 (2023): 435.
- [10] Stein, Hannah, et al. "Production of isolated giant unilamellar vesicles under high salt concentrations." *Frontiers in physiology* 8 (2017): 63.

- To be consistent with DOPC and DOPS, use SOPC for 18:0-18:1 PC and POPE for 16:0-18:1 PE. And why do the authors chose to prepare SOPC, then POPE and for the 1:1 mixture used DOPC:DOPE? What's the explanation?

[RESPONSE] We appreciate the reviewer's meticulous observation. Following the reviewer's suggestion, we have made the necessary changes in the manuscript to use SOPC for 18:0-18:1 PC and POPE for 16:0-18:1 PE to maintain consistency with DOPC and DOPS.

Regarding the selection of different lipid compositions in our study, we aimed to explore the applicability of our method to various lipid compositions within the constraints of the manuscript's length. To demonstrate the versatility of our approach, we utilized different lipid types, such as SOPC, POPE, and DOPC, both individually and in combination, to fabricate 3DFLBs. This diverse lipid composition allowed us to showcase the broad applicability of our method across different lipid compositions. Therefore, the selection of SOPC, POPE, and the 1:1 mixture of DOPC:DOPE was strategic, enabling us to demonstrate the method's effectiveness across a range of lipid compositions.

- line 203: DOPC:DOPE:DOP=50:45:5 add the "S" of DOPS. Also, as mentioned above, 5mol% DOPS is not enough to characterize a "charged" system. Did the authors tried to grow compositions with higher charge, like 30-50%, and where not successful?

[RESPONSE] We appreciate the reviewer's meticulous attention to detail in identifying our typo. Following the comment, we have rectified the omission of "S" in DOPS and updated the manuscript accordingly.

As mentioned in our previous response, we introduced hydraulic pressure to regulate membrane fusion via electroformation in physiological solutions and successfully formed and sealed 3DFLBs using a lipid mixture containing 30 mol% DOPS. We have accordingly replaced the data in Figure 31 with a new one.

Additionally, we continue efforts to generate 3DFLBs containing 40 mol% DOPS. However, when we increased the proportion of DOPS to 40 mol%, we were able to generate 3DFLBs but encountered difficulties in achieving tight sealing (as illustrated below). We found that generating 3DFLBs with lipid mixtures containing more than 40 mol% DOPS required hydraulic pressure beyond what was mentioned in the manuscript, and we believe that

further efforts could lead to the successful generation and sealing of 3DFLBs with 40 mol% DOPS.

Nevertheless, it is worth noting, as mentioned in our previous response, that the proportion of charged lipids typically does not exceed 20 mol% in the field of lipid membrane fabrication via electroformation. Producing lipid membranes with a higher proportion of charged lipids via electroformation presents a significant challenge and demands substantial effort in this field. We hope our continuous efforts bring enhancement to this field and will report in the following paper.

[MODIFICATION] We modified some sentences and figures to the manuscript as below

Modified sentence in the “Results” section

“... and even a mixture containing charged lipid (DOPC:DOPE:DOP=50:45:5; Fig. 31) ...”

to

“... and even a mixture containing charged lipid (DOPC:DOPE:DOPS=50:20:30; Fig. 31) ...”

- There are still mentions to “SUV solution” (lines 321,322). Change to “dispersion”.

[RESPONSE] Thank you for the meticulous observation and we very sorry for our missing. We revised “SUV solution” to “SUV dispersion” in the “Patterning of lipids in the microwell array” section (lines 321, 322).

- What is the meaning of “Reconstitution” of melittin on the 3DFLBs? This term is generally applied for incorporation of a membrane protein into a bilayer structure for instance, and not to the lytic action of a membrane-active peptide such as melittin.

[RESPONSE] Thank you for your thoughtful comment. We agree with the reviewer's point that "Reconstitution" is typically used for the incorporation of membrane proteins into a bilayer structure. This was indeed an oversight on our part. We have rectified this by updating "Reconstitution" to "Incorporation" in the "Method" section of the Manuscript.

- line 391: “To observe this fluorescence effect in vesicles other than the 3DFLBs”, what fluorescence effect is it referring to, since this is the first sentence in the section?

[RESPONSE] Thank you for your careful comment. There was a typo which raised confusing. We revised the sentence as below.

[MODIFICATION] Modified sentence in the “Method” section.

“To observe this fluorescence effect in vesicles other than the 3DFLBs ...”

to

“To observe the fluorescence effect in vesicles other than the 3DFLBs ...”

- Figure S1: Change “Lipid solution” for “SUVs” or “SUVs dispersion”. Lipid solution gives the impression of lipid dissolved in chloroform. On the second snapshot, why are the SUVs of such different sizes, as they show a quite narrow size distribution? Explain in the caption what “DI” water mean.

[RESPONSE] Thank you for the reviewer's comment. Regarding Figure S1, there is no mention of "Lipid solution." We believe "Lipid solution" may pertain to Figure S3. We have accordingly amended "Lipid solution" in Figure S3 to "SUVs dispersion."

As for the different sizes of SUVs in the second snapshot, this was intended to represent the size distribution. However, to avoid any confusion, we have adjusted the representation to accurately reflect the narrow size distribution of the SUVs.

Regarding the term "DI" water, it stands for deionized water. While the "Method" section in the Manuscript provides an explanation for this abbreviation, to prevent any misunderstanding, we have revised "DI water" to "deionized water" in Figure S3.

- According to the journal guidelines “The main text of an Article should begin with a section headed Introduction of referenced text that expands on the background of the work (some overlap with the abstract is acceptable), followed by sections headed Results, Discussion (if appropriate) and Methods (if appropriate).” So please add the section “Introduction”.

[RESPONSE] We appreciate the reviewer’s precious comment. Following the reviewer's comment and in adherence to the journal guidelines, we have incorporated a section titled "Introduction" into the manuscript.

Response to the reviewer 3's comments

I appreciate the authors' efforts in elucidating the obtained membrane structures in more detail, which significantly helped with the readability of the manuscript.

However, my main concern is the fact that the resulting membrane structures are very similar to those previously reported by the group and this concern questions the significance and novelty of this work. In comparison to their prior research, where they operated at a hydraulic pressure of 0 kPa in sucrose, the current study explored again the influence of AC frequencies but this time in conjunction with hydraulic pressure, which enabled them to use not only sucrose but salt. This expansion has led to the discovery of a regime enabling the production of tightly sealed 3DFLBs in the presence of salt, offering the potential replacement of sucrose – a noteworthy advancement for protein reconstitution.

[RESPONSE] We deeply appreciate the thoughtful feedback provided by the reviewer. While certain aspects of our current study bear similarities to our previous research, it's essential to highlight the significant achievement we've made in this study: **the formation of highly stable 3DFLBs through precise control of membrane fusion in physiological solutions.** This represents a notable advancement beyond conventional methods. Our paper aims to overcome these limitations and venture into device applications by creating stable 3DFLBs on solid substrates as an array form, utilizing lipid structures of various shapes. The novelty of our study lies in providing a solution for fabricating stable 3D lipid bilayer structures through precise control of membrane fusion via pressure and electric field manipulation. Additionally, 3DFLBs hold promise for future electrophysiological measurements of ion transports.

While acknowledging this progress, it's important to recognize that this achievement requires the independent control of an additional parameter, necessitating a more intricate setup. The membrane structures unveiled may pave the way for future experiments, yet their specific applications have not been demonstrated in this study, leaving the practical gain in knowledge somewhat moderate. A clear advancement of the current membrane structures would be the reconstitution of a transmembrane protein, which is known to be still a challenge for the community working with large vesicle structures such as GUVs. The authors used only melittin, a soluble peptide, which was a replacement for hemolysin (as they state in their response) and that they applied to show the unilamellarity of the 3DFLBs.

[RESPONSE] To address the reviewer's concern regarding the creation of lipid bilayers incorporating membrane proteins, we introduced a straightforward approach using hydraulic pressure to fuse small unilamellar vesicles (SUVs) to 3DFLBs. This method demonstrates the potential for recombining SUVs or proteoliposomes containing various cell components into 3DFLBs, indicating its utility in constructing artificial cell membranes. While various protocols exist for creating micrometer-sized vesicles containing membrane proteins, they often suffer from low efficiency and variability. Reconstituting purified membrane proteins into such vesicles remains a challenging task. One approach involves creating membrane protein-embedded small vesicles tailored to each protein type, which can then be inserted into target large membranes through fusion. In our previous study, we addressed the challenge of membrane fusion using charged lipids to increase the fusion rate and successfully reconstituted the 5HT-3A (serotonin receptor) into 3DFLBs, confirming protein activity. In this paper, we've elevated the fusion rate between small unilamellar vesicles and 3DFLBs using hydraulic pressure, as demonstrated in Figures 4d-f. This method, simpler and more universal compared to our previous approach via charged lipids, holds promise for reassembling various cell components, including membrane proteins, into artificial cell membranes.

In conclusion, our study aims to overcome the limitations of conventional electroformation by regulating membrane fusion and fabricating solvent-free freestanding lipid bilayers. The generated 3DFLBs exhibit remarkable stability and broad applicability. While we acknowledge the reviewer's valid concern regarding lipid mixtures with integral membrane proteins, addressing this matter extends beyond the scope of our current work and warrants further exploration in future studies.

REVIEWERS' COMMENTS

Reviewer #2 (Remarks to the Author):

The authors addressed all my comments except the distinction between SUVs and LUVs. Even though I think it would make much more sense to call SUVs those obtained through 50nm-pores and LUVs those through 100nm, I can let go of this minor issue.

Reviewer #3 (Remarks to the Author):

Although I do not yet fully agree with the authors that the results represent a major advance in the general technique, the future will show whether the method will become widely applicable and whether it will pave the way for routine incorporation of transmembrane proteins. I have no further comments.

Response to the reviewer 2's comments

The authors addressed all my comments except the distinction between SUVs and LUVs. Even though I think it would make much more sense to call SUVs those obtained through 50nm-pores and LUVs those through 100nm, I can let go of this minor issue.

[RESPONSE] We appreciate the reviewer's feedback and have made the necessary adjustments to the manuscript. The vesicles obtained through the 50 nm pores are now referred to as LUVs instead of SUVs, as suggested.

Response to the reviewer 3's comments

Although I do not yet fully agree with the authors that the results represent a major advance in the general technique, the future will show whether the method will become widely applicable and whether it will pave the way for routine incorporation of transmembrane proteins. I have no further comments.

[RESPONSE] We appreciate the reviewer's thoughtful consideration of our work. We expect that the true impact and applicability of the method will become clearer with further research and its potential for incorporation of transmembrane proteins.